# Programmed genome editing of the omega-1 ribonuclease of the blood fluke, *Schistosoma mansoni*

Wannaporn Ittiprasert[1,2], Victoria H Mann[1,2], Shannon E Karinshak[2], Avril Coghlan[3], Gabriel Rinaldi[3], Geetha Sankaranarayanan[3], Apisit Chaidee[2,4], Toshihiko Tanno[5,6], Chutima Kumkhaek[7], Pannathee Prangtaworn[2,8], Margaret M Mentink-Kane[9], Christina J Cochran[1], Patrick Driguez[3], Nancy Holroyd[3], Alan Tracey[3], Rutchanee Rodpai[4], Bart Everts[10], Cornelis H Hokke[10], Karl F Hoffmann[11], Matthew Berriman[3], Paul J Brindley[1]*

[1]Department of Microbiology, Immunology and Tropical Medicine, School of Medicine and Health Sciences, George Washington University, Washington, DC, United States; [2]Research Center for Neglected Diseases of Poverty, School of Medicine and Health Sciences, George Washington University, Washington, DC, United States; [3]Wellcome Sanger Institute, Wellcome Genome Campus, Hinxton, United Kingdom; [4]Department of Parasitology, Faculty of Medicine, Khon Kaen University, Khon Kaen, Thailand; [5]Department of Surgery, University of Maryland, Baltimore, United States; [6]Institute of Human Virology, University of Maryland, Baltimore, United States; [7]Cellular and Molecular Therapeutics Laboratory, National Heart, Lungs and Blood Institute, National Institutes of Health, Bethesda, United States; [8]Department of Parasitology, Faculty of Medicine Siriraj Hospital, Mahidol University, Bangkok, Thailand; [9]Schistosomiasis Resource Center, Biomedical Research Institute, Rockville, United States; [10]Department of Parasitology, Leiden University Medical Center, Leiden, Netherlands; [11]Institute of Biological, Environmental and Rural Sciences, Aberystwyth University, Aberystwyth, United Kingdom

*For correspondence: pbrindley@gwu.edu

Competing interests: The authors declare that no competing interests exist.

**Abstract** CRISPR/Cas9-based genome editing has yet to be reported in species of the Platyhelminthes. We tested this approach by targeting omega-1 (ω1) of *Schistosoma mansoni* as proof of principle. This secreted ribonuclease is crucial for Th2 polarization and granuloma formation. Schistosome eggs were exposed to Cas9 complexed with guide RNA complementary to ω1 by electroporation or by transduction with lentiviral particles. Some eggs were also transfected with a single stranded donor template. Sequences of amplicons from gene-edited parasites exhibited Cas9-catalyzed mutations including homology directed repaired alleles, and other analyses revealed depletion of ω1 transcripts and the ribonuclease. Gene-edited eggs failed to polarize Th2 cytokine responses in macrophage/T-cell co-cultures, while the volume of pulmonary granulomas surrounding ω1-mutated eggs following tail-vein injection into mice was vastly reduced. Knock-out of ω1 and the diminished levels of these cytokines following exposure showcase the novel application of programmed gene editing for functional genomics in schistosomes.
DOI: https://doi.org/10.7554/eLife.41337.001

**eLife digest** Schistosomiasis is a tropical disease that can cause serious health problems, including damage to the liver and kidneys, infertility and bladder cancer. Nearly a quarter billion people are currently infected, mostly in poor regions of sub-Saharan Africa, the Philippines and Brazil.

A freshwater worm known as *Schistosoma mansoni* causes the disease. These parasites enter the human body by burrowing into the skin; once in the bloodstream, they move to various organs where they rapidly start to reproduce. Their eggs release several molecules, including a protein known as omega-1 ribonuclease, which can damage the surrounding tissues.

A gene editing technique called CRISPR/Cas9 allows scientists to precisely target and then deactivate the genetic information a cell needs to produce a given protein. While the tool has been used in other species before, it was unknown if it could be applied to *S. mansoni*. Here, Ittiprasert et al. harnessed CRISPR/Cas9 to deactivate the gene that codes for omega-1 ribonuclease and create parasites that do not produce the protein, or only very little of it. The experiments showed that mice infected with the gene-edited worm eggs displayed far fewer symptoms of schistosomiasis compared to those that carry the non-edited parasites.

Alongside this work, Arunsan et al. used CRISPR/Cas9 to inactivate a gene in another species of worm that can cause liver cancer in humans. Together, these findings demonstrate for the first time that the gene editing method can be adapted for use in parasitic flatworms, which are a major public health problem in tropical climates. This tool should help scientists understand how the parasites invade and damage our bodies, and provide new ideas for treatment and disease control.

DOI: https://doi.org/10.7554/eLife.41337.002

## Introduction

Schistosomiasis is considered the most virulent of the human helminth diseases in terms of morbidity and mortality (*Gryseels et al., 2006*; *Hotez, 2014a*; *Hotez, 2014b*; *Hotez et al., 2008*; *Colley, 2014*; *Colley et al., 2014*). The past decade has seen major advances in knowledge and understanding of the pathophysiology, developmental biology, evolutionary relationships and genome annotation of the human schistosomes (*Berriman et al., 2009*; *Young et al., 2012*; *Lepesant et al., 2011*; *Geyer et al., 2011*; *Vanderstraete et al., 2014*; *Rinaldi et al., 2012a*; *Rinaldi et al., 2012b*; *Protasio et al., 2012*; *Valentim et al., 2013*; *Wang et al., 2013*; *Collins et al., 2013*; *Hagen et al., 2014*). Establishing CRISPR/Cas9 genome editing in schistosomiasis would greatly enable effective functional genomics approaches. Stable CRISPR/Cas9-based site-specific gene mutation and phenotyping will drive innovation and a deeper understanding of schistosome pathogenesis, biology and evolution (*Hoffmann et al., 2014*).

The schistosome egg plays a central role in disease pathogenesis, causation and transmission (*Gryseels et al., 2006*). The appearance of *S. mansoni* eggs in host tissues by 6 to 7 weeks after infection coincides with profound polarization to a granulomatous, T helper type 2 (Th2) cell phenotype (*Pearce et al., 2012*; *Everts et al., 2012*; *Fairfax et al., 2012*; *Steinfelder et al., 2009*; *Everts et al., 2009*; *Pearce et al., 2004*). Numerous egg proteins have been characterized, with >1000 identified in a well-studied fraction termed soluble egg antigen (SEA) (*Dunne et al., 1991*; *Curwen et al., 2004*; *Ashton et al., 2001*; *Cass et al., 2007*; *Mathieson and Wilson, 2010*). In viable eggs, about 30 of the SEA proteins are located outside the developing miracidium and encompass the complement of secreted antigens (egg-secreted proteins, ESP) that interact with host tissues to facilitate the passage of the egg from the mesenteric veins to the intestinal lumen (*Mathieson and Wilson, 2010*). The T2 ribonuclease omega-1 (ω1) is the principal Th2-inducing component of ESP with its Th2-polarizing activity dependent upon both its RNase activity and glycosylation (*Steinfelder et al., 2009*; *Everts et al., 2009*; *Wilbers et al., 2017*). This RNase is hepatotoxic (*Fitzsimmons et al., 2005*), and its secretion by eggs into the granuloma regulates the pattern recognition receptor signaling pathways in dendritic cells that, in turn, prime Th2 responses from CD4[+] T cells (*Ferguson et al., 2015*). Secreted ω1 provokes granulomatous inflammation around eggs traversing the wall of the intestines, and trapped in hepatic sinusoids and other host organs,

driving fibrosis that eventually results in hepatointestinal schistosomiasis (*Gryseels et al., 2006*; *Wynn et al., 2004*).

As ω1 drives distinctive immunological phenotypes including Th2 polarization and granuloma formation, we investigated the use of programmed CRISPR/Cas9-mediated genome editing (*Jinek et al., 2012*; *Hsu et al., 2014*) to alter the ω1 locus by both gene knockout and knock-in approaches. The investigation revealed that programmable genome editing catalyzed by the bacterial endonuclease Cas9 was active in schistosomes, with chromosomal double stranded breaks (DSB) repaired by homology directed repair (HDR) using a donor, single-stranded oligonucleotide template bearing short homology arms and/or by non-homologous end joining (NHEJ). The programmed mutagenesis decreased levels of ω1 mRNA and induced distinct *in vitro* and *in vivo* phenotypes, including a substantial loss of capacity of SEA from ω1-mutated eggs to polarize Th2 cytokine responses (IL-4 and IL-5) in co-cultured macrophages and T cells and loss of capacity to provoke formation of pulmonary granulomas *in vivo*. Functional knock-out of ω1 and the resulting immunologically impaired phenotype showcase the novel application of CRISPR/Cas9 and its utility for functional genomics in schistosomes.

## Results

### Omega-1, a multicopy locus on chromosome 1 of *S. mansoni* specifically expressed in eggs

Five genomic copies of ω1 were identified in the *S. mansoni* reference genome, version 5 (*Figure 1—figure supplement 1*), although the repetitive nature of the ω1 locus on chromosome one presents a challenge for genome assembly. A single copy of ω1 selected for genome editing, Smp_193860, included nine exons separated by eight introns and spanned 6,196 nt (*Figure 1A*). Several other copies shared similar exon/intron structure and conserved coding sequences (*Figure 1—figure supplement 1A and B*). The predicted coding sequence (CDS) of Smp_193860 encoded a ~27 kDa protein, of similar mass to the 31 kDa reported for ω1 (*Murare et al., 1992*). The gene encodes a secreted ribonuclease of the T2 family of transferase-type endoribonucleases with conserved catalytic regions (*Luhtala and Parker, 2010*). We designed a sgRNA targeting residues 3,808 to 3,827 of Smp_193860 within exon 6, adjacent to an AGG protospacer adjacent motif (PAM) and with the predicted Cas9 cleavage site located at three residues upstream of the PAM. *Figure 2—figure supplement 1* provides the nucleotide sequence of the Smp_193860 copy, and indicates the UTR, coding exons and introns; 6,196 nt. The AGG and the nucleotide sequence complementary to this sgRNA were also present in the Smp_179960 and Smp_184360 copies of ω1. These three copies shared >99% identity in the 202 bp PCR amplicon region subjected to next generation sequencing (NGS) although they differed by several substitutions (*Figure 2—figure supplement 2A,B and C*). All three copies of ω1 display a tight profile of developmental stage expression: expression is restricted to the mature egg with expression not apparent elsewhere during the developmental cycle of this schistosome (*Figure 1—figure supplement 1C*) (*Lu et al., 2018*).

### Homology directed repair and non-homologous end joining pathways in schistosomes

The draft genome of *S. mansoni* was surveyed for key proteins of the non-homologous end joining (NHEJ) and homology-directed repair (HDR) pathways. Artemis and DNA-PKcs are essential NHEJ factors in vertebrates (*Deriano and Roth, 2013*; *Lee et al., 2014*; *Pardo et al., 2009*). Candidate homologues for six of seven human NHEJ pathway genes and for two key HDR pathway genes were identified by searching for matches to Pfam (*Supplementary file 1*). A putative homologue of Cernunnos/XLF was not apparent in *S. mansoni* (*Deriano and Roth, 2013*) based on searching for the Pfam XLF domain (PF09302) found in human Cernunnos/XLF. The domain appears to be absent from all flatworm species studied by the International Helminth Genomes Consortium (*International Helminth Genomes Consortium, 2019*).

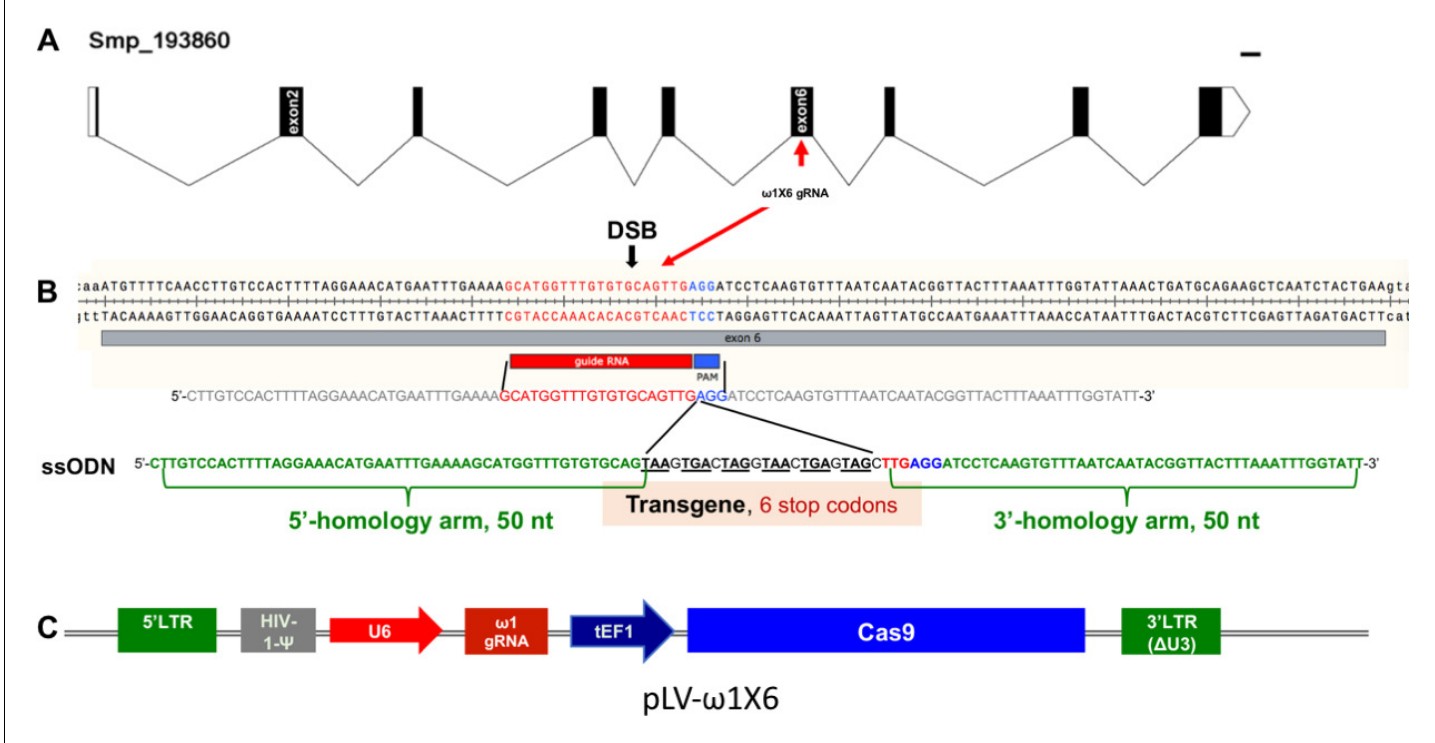

**Figure 1.** Genomic structure of the locus encoding omega-1 (ω1) in the genome of *S.mansoni,* guide RNA and CRISPR/Cas9 encoding construct. (A) Gene model of ω1 (Smp_193860), showing the position of its nine exons, eight introns and UTRs 6,196 bp on chromosome 1. (B) Nucleotide sequence in exon six indicating location and sequence of gRNA target site, predicted double-stranded break (DSB) (arrow), protospacer adjacent motif (PAM) (AGG, blue box), and 124-nucleotide sequence of the single-stranded DNA donor template provided for DSB repair by homologous recombination. Homology arms of 50 nt flank a central 24 nt of six-stop-codon transgene. (C) Linear map of pLV-ω1X6 showing position of regulatory and coding regions for CRISPR/Cas9 editing; the positions of the human U6 promoter to drive ω1 gRNA, the translational elongation factor EF1-α promoter driving Cas9 from *S. pyogenes*, and the left and right long terminal repeats of the lentiviral vector derived from HIV-1.
DOI: https://doi.org/10.7554/eLife.41337.003

The following figure supplement is available for figure 1:

**Figure supplement 1.** The T2 ribonuclease ω1 is encoded by at least five gene copies located on *S.*
DOI: https://doi.org/10.7554/eLife.41337.004

## Site-specific integration of transgene confirmed CRISPR-Cas9 activity in schistosomes

The activity and efficiency of CRISPR/Cas9 to edit the schistosome genome, by targeting the ω1 locus, was explored using two approaches. First, a ribonucleoprotein complex (RNP) comprising of sgRNA mixed with recombinant Cas9 endonuclease was delivered into schistosome eggs isolated from livers of experimentally infected mice (eggs termed 'LE', for 'liver eggs') by electroporation. In addition, homology directed repair (HDR) of CRISPR/Cas9-induced double stranded breaks (DSBs) at ω1 in the presence of a donor DNA template was investigated (*Lok et al., 2017*; *Gang et al., 2017*; *Chen et al., 2015*). A single-stranded oligodeoxynucleotide (ssODN) of 124 nt in length was delivered to some LE as a template for HDR of chromosomal DSBs (*Figure 1B*). The ssODN included a short transgene encoding six stop codons flanked by 5'- and 3'-homology arms, each arm 50 nt in length, complementary to the genome sequence of exon 6 on the 5´ and 3´ sides of the Cas9 cleavage site (*Figure 1A and B*). In a second approach, a lentivirus vector (pLV-ω1X6; *Figure 1A and B*) that included Cas9, driven by the mammalian translational elongation factor one promoter, and the exon 6-targeting sgRNA (20 nt), driven by the human U6 promoter (*Duvoisin et al., 2012*) was engineered. LE were transduced with pseudotyped lentiviral virions (pLV) by exposure in culture to LE for 24 hr (*Rinaldi et al., 2012a*; *Suttiprapa et al., 2016*) and, thereafter, transfected with the ssODN repair template. In both approaches, expression of ω1 in LE after 72 hr in culture was ascertained.

Given that the donor ssODN included a short transgene that facilitates genotyping, PCR was performed using template genomic DNAs from the CRISPR/Cas9-treated LE (*Lok et al., 2017*) to reveal the site-specific knock-in (KI). A forward primer termed Smω1X6–6 stp-cds-F specific for the ssODN transgene was paired with three discrete reverse primers, termed Smω1-R1, Smω1-R2 and Smω1-R3, at increasing distance from the predicted HDR insertion site in ω1 (*Supplementary file 2*). Amplicons of the expected sizes of 184, 285 and 321 nt were observed in genome-edited eggs but not in control eggs (*Figure 1A and B*, *Figure 2—figure supplement 3A*), a diagnostic pattern indicative of the ssODN transgene insertion into ω1 and, in turn, indicating that the resolution of the DSB at the ω1 locus from CRISPR/Cas9 had been mediated by HDR. Amplification using a control primer pair that spanned the predicted DSB, termed Smω1-control-F/R, yielded control amplicons of the expected 991 nt. Similar findings were observed with genome editing delivered by RNPs and by LV (*Figure 2—figure supplement 3*). Sanger sequence analysis of the knocked-in amplicons (KI-R1, KI-R2 and KI-R3) confirmed the presence of the transgene inserted into ω1 at the site targeted for programmed cleavage (*Figure 2C*).

## Programmed mutations in exon 6 of ϖ1

The activity of CRISPR/Cas9 was first evaluated by a quantitative PCR (qPCR) approach that relies on the inefficient binding of a primer overlapping the gRNA target, which is where mutations were expected to have occurred, compared to the binding efficiency of flanking primers, which were outside the mutated region (*Shah et al., 2015*; *Yu et al., 2014*). The overlapping (OVR) primer pair shared the reverse primer with OUT primer (*Figure 3—figure supplement 1A*). Genomic DNA template was used for qPCR to quantify the efficiency of CRISPR-mediated mutagenesis at the target locus; the ratio between the OVR products and OUT products estimate the relative fold amplification reduction in CRISPR/Cas9-manipulated samples compared to controls in the target sequence of the gRNA. Relative fold amplification was reduced by 12.5% in gDNA isolated from eggs treated with pLV-ω1X6 and ssODN, whereas a reduction in relative fold amplification of 2.5, 6.9, and 4.5 were observed in eggs treated with gRNA/Cas9 RNP complex alone, gRNA/Cas9 RNP complex and ssODN, or pLV-ω1X6 alone, respectively. A reduction in relative fold amplification was not apparent among control groups, that is untreated eggs, eggs electroporated in the presence of Opti-MEM only, Cas9 only, eggs transduced with heat-inactivated pLV-ω1X6 with ssODN donor, and eggs transfected with ssODN only (*Figure 3—figure supplement 1B*).

To further characterize and quantify the mutations that arose in the genome of ω1 gene-edited eggs, we used an amplicon next generation sequencing approach. Barcoded amplicon libraries were constructed from pooled genomic DNA of six independent exposures of LE to pLV-ω1X6 and the donor ssODN. Each amplicon was sequenced on the MiSeq Illumina platform and the CRISPResso pipeline (http://crispresso.rocks/) (*Pinello et al., 2016*; *Canver et al., 2018*) was used to analyze deep-coverage sequence reads. More than 56 million sequenced reads were compared to the reference amplicon sequence of the Smp_193860 locus (*Supplementary file 3*), which revealed that 71% exhibited the wild type (WT; *i.e.,* unmodified DNA) whereas 29% reads exhibited apparent mutations (*Figure 2D*) across the 202 bp amplicon, with 0.13% insertions, 0.58% deletions and 28.2% substitutions (*Supplementary file 3*, sample 9). In addition, the deletions in treated samples compared to controls are longer around the DSB predicted site (*Figure 2D*). In contrast, in the control eggs-only group, 76% were WT, and 24% of reads exhibited apparent mutations, with 0.14% insertions, 0.33% deletions, and 24.0% substitutions (*Figure 2E*, sample two in *Supplementary file 3*). Thus, subtracting the rate of apparent mutations in the control, we estimated that 0.25% and 4.2% of reads in the experimental sample carried programmed CRISPR-induced deletions and substitutions, respectively. Indels of 1–2 bp, or multiples thereof, in coding DNA cause frame-shifts, and consistent with its higher rate of indels, the CRISPR/Cas9-treated sample displayed a higher rate of frame-shifts compared to a sample from control eggs, 2.0% versus 1.4%).

Many apparent sequence variants common to the control and edited eggs likely reflect polymorphism among copies of ω1 rather than programmed mutations. The sequence reads revealed several common variants, such as adjacent 'TA' substitutions instead of 'CC' at positions 152–153 of the amplicon, which encodes a change from K to Q at the amino acid level. The gene Smp_193860 has 'TA' at this position in the V5 assembly (*Protasio et al., 2012*), as does the mRNA XP_018647487.1 from the NCBI database, whereas Smp_193860, Smp_184360 and Smp_179960 all have 'CC' at this position in the V7 assembly (Berriman and co-workers, in preparation) (*Figure 2—figure supplement*

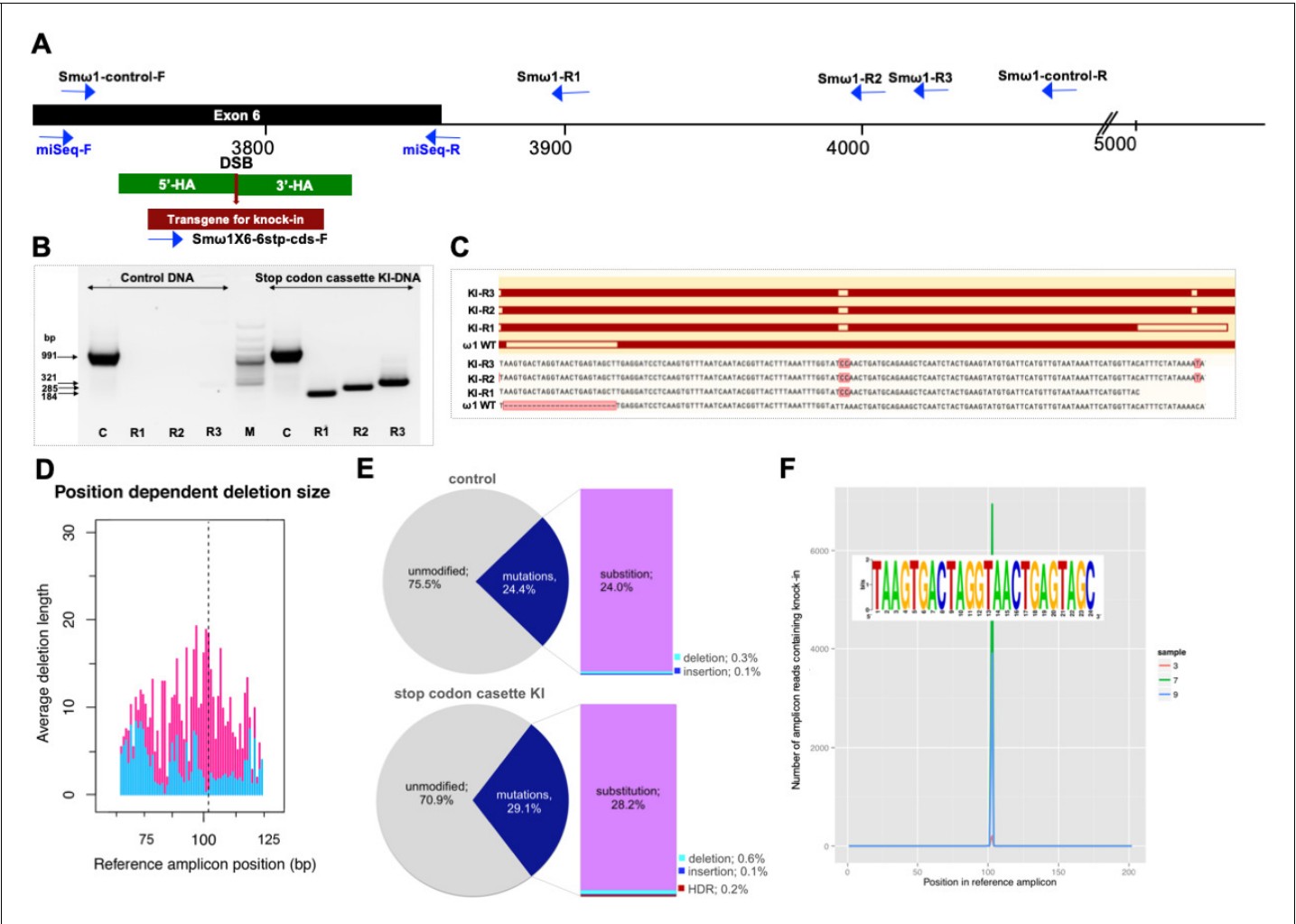

**Figure 2.** Programmed chromosomal break at ω1 locus repaired by homologous recombination from donor template. (**A**) Schematic diagram to indicate positions of primer binding sites (blue arrows), with the foreign gene cassette as the forward primer (Smω1X6–6 stop codons cassette-F) paired with three discrete reverse primers, Smω1-R1, Smω1-R2 and Smω1-R3 from the ω1 locus and a primer pair for target amplicon NGS library amplification; miSeq-F and miSeq-R. The control PCR amplicon was generated using the Smω1-control-F and –R primers. The green box shows the location of 5' and 3' homology arms, the red box and arrow indicate the stop codon bearing transgene. (**B**) PCR products visualized in ethidium bromide-stained agarose gel demonstrating Cas9-catalyzed target site-specific insertional mutagenesis in exon 6 of the ω1 gene. Evidence for transgene knocked-in into programmed target site revealed by amplicons of the expected sizes in lanes R1, R2 and R3, of 184, 285 and 321 bp, respectively (arrows at left) spanned the mutated site in the genomic DNAs pooled from schistosome eggs, including a positive control flanking the insert site (991 bp). The control DNA result shown in this gel was isolated from heat-inactivated-pLV-ω1X6 virions and ssODN treated LE. Similar findings were obtained when programmed gene editing was executed by lentiviral virion-delivered Cas9 and ω1-gRNA transgenes and by ribonucleoprotein complex (RNP) delivered by square wave electroporation (supporting information). The non-KI control groups (sgRNA only, heat-inactivated pLV-ω1X6 virions only, ssODN only) showed no amplicons by stop cassette-KI primers with R1, R2 or R3. (**C**) Multiple sequence alignments confirmed the presence of the 24 nt transgene inserted precisely into exon 6 of ω1 locus from KI-R1, -R2 and -R3 fragments compared with ω1 wild type (WT). The white box on ω1-WT indicates the absence of the transgene sequence and white boxes on KI-R1, -R2 and –R3 fragments show locations of substitutions relative to the other ω1 copies (Smp_184360): 2 bp (AT to CC) mismatches at positions 253–254 nt. All three contained the (knock-in) insertion sequence (white box), which confirmed targeted mutation of the ω1 gene. (**D–F**) Illumina deep sequence analysis of amplicon libraries revealed Cas9 induced on-target repair of programmed gene mutation of the ω1 locus by deletions, insertions, and substitutions by CRISPResso analysis. *D*; position dependent deletion size (deletion site, X-axis; deletion size, Y-axis); the deletions varied in length from point mutations to >20 bp adjacent to the DSB. The dotted line indicates the predicted position of the programmed double-stranded break. (**E**), frequency of frameshift versus in-frame mutations reported by CRISPResso. The pie charts show the fraction of all mutations (indels and substitutions) in the coding region (positions 42–179) of the amplicon predicted to induce frameshifts, that is indels of 1–2 bp, or multiples thereof. Top graph corresponds to sample 2 (eggs only control) (*Supplementary file 3*), bottom graph corresponds to sample 9 (eggs exposed to virions and ssODN, that is CRISPR/Cas9-treated) (*Supplementary file 3*). Findings for control and treated samples are provided in *Supplementary File 3*. (**F**), Frequency distribution of insertions of the knock-in transgene. Number of amplicon reads containing an insertion of the knock-in sequence (with ≥75% identity to it) is shown in the Y-axis, and

*Figure 2 continued on next page*

*Figure 2 continued*

the position of the insertion relative to the reference amplicon is shown on the X-axis. The programmed Cas9 scission lies between positions 102 and 103. Samples 3, 7 and 9 are independent amplicon libraries (technical replicates) made from the same sample of genomic DNA pooled from six biological replicates exposed to virions and ssODN. The insert shows a sequence logo, created using WebLogo (*Crooks et al., 2004*), of the sequences of the 3826 sequence reads from samples 3, 7 and 9, with insertions of 24 bp at position 102; most matched the donor template, TAAGTGACTAGGTAACTGAGTAGC.

DOI: https://doi.org/10.7554/eLife.41337.005

The following figure supplements are available for figure 2:

**Figure supplement 1.** The nucleotide sequence of the Smp_193860 copy, and indicates the UTR (green), coding exons (blue) and introns (red).

DOI: https://doi.org/10.7554/eLife.41337.006

**Figure supplement 2.** Variation among copies of ω1 in the reference genome of *S.mansoni*.

DOI: https://doi.org/10.7554/eLife.41337.007

**Figure supplement 3.** Programmed HDR-catalyzed knock-in of transgene into ω1, exon 6.

DOI: https://doi.org/10.7554/eLife.41337.008

**Figure supplement 4.** Estimated relative copy number of ω1.

DOI: https://doi.org/10.7554/eLife.41337.009

*2B and C*). In addition, 'CC' was also observed in KI fragments by Sanger direct sequencing (*Figure 2C*). A second common dinucleotide substitution from 'AC' to 'TT' at positions 60–61 encodes an amino acid change from T to F. Both dinucleotide substitutions occurred together in 8% of reads in the control group (*Supplementary file 3*, sample 2) and 4% of reads in the gene-edited group (*Supplementary file 3*, sample 9). These non-synonymous substitutions may have functional significance given their proximity to the catalytic site of the ribonuclease (*Figure 2—figure supplement 2C*).

Along with the predicted NHEJ-catalyzed mutations, CRISPResso (http://crispresso.rocks/) determined the rate of HDR-mediated ssODN knock-in (*Figure 2F* and *Supplementary file 3*). Here, insertion of the 24 bp transgene was confirmed in 0.19% of the reads at the sgRNA programmed CRISPR/Cas9 target site (*Figure 2F*, sample nine in *Supplementary file 3*). Some reads containing the knock-in sequence included the 'CC' to 'TA' substitutions at positions 152–153 and 'AC' to 'TT' at positions 60–61 (*Figure 2—figure supplement 2*). This indicates that the indels catalyzed by NHEJ and/or KI by HDR occurred in target DNA sequences which exhibited 99% identity in multiple copies of ω1 including Smp_193860, Smp_184360 and Smp_179960, and possibly also further copies not yet annotated in the reference genome. The qPCR approach estimated a reduction by 12.5% in relative fold amplification in the pLV-ω1X6 with ssODN treatment group (*Figure 3—figure supplement 1*), whereas CRISPResso analysis of the pooled NGS reads indicated a frequency of indel/substitution mutation of ~4.5% (*Supplementary file 3*).

## Programmed gene editing markedly diminished the expression of ω1

Liver eggs (LE) transfected with RNP complexes, with or without ssODN, displayed a downregulation of the ω1-specific transcript of ~45% and 81%, respectively, compared to controls (p≤0.05; n = 11 by one way ANOVA). However, LE transduced with pLV-ω1X6 virions, with or without ssODN, showed a reduction of the ω1-specific transcripts of 67% and 83% respectively, when compared to controls (*Figure 3A*). Similar outcomes were seen in all biological replicates undertaken (n = 11). This outcome indicated that resolution of chromosomal DSB by NHEJ plus HDR provided enhanced programmed gene knockout compared to NHEJ-mediated chromosomal repair alone. Nevertheless, both RNPs and pLV virions efficiently delivered programmed gene editing to schistosomes but lentiviral transduction delivered enhanced performance with stronger gene silencing, in particular when the donor repair template was provided (*Figure 3A*). When examined at later time points (days 5 and 7 following manipulation of the LE), further reduction in ω1 abundance was not apparent (*Figure 3B*).

Large DNA deletions have been associated with CRISPR/Cas9 mutations in another helminth species (*Gang et al., 2017*). However, using qPCR to estimate relative copy number, as previously described (*Suttiprapa et al., 2012a*), there were no evidence that silencing of ω1 was associated with the ω1 multiple gene copy numbers (*Figure 2—figure supplement 4*).

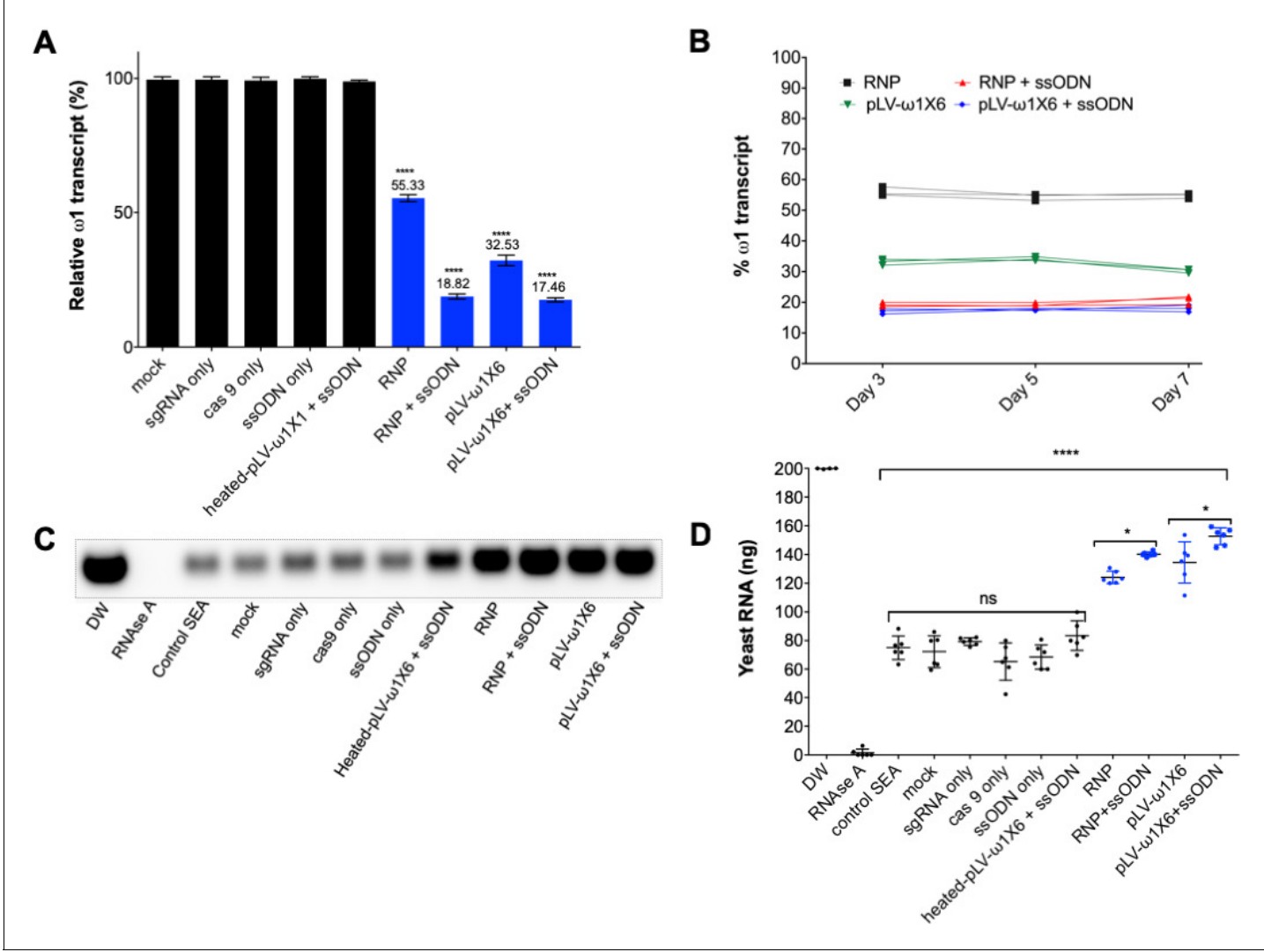

**Figure 3.** Diminished ω1-specific transcript levels and ribonuclease T2 activity following programmed editing. (A) ω1 mRNA abundance reduction up to ~70% after genome editing by sgRNA/Cas9 complex and lentivirus systems, and markedly reduced >80% with the addition of ssODN as the DNA repair donor. Relative expression of ω1 transcripts at 3 days following CRISPR/Cas9 manipulation; mean ±SD, n = 11 (biological replicates); p≤0.0001 (****) ANOVA statistic significant. (B) Stable reduction of ω1 transcripts at days 5 and 7 after treatment (three biological replicates) in four experimental groups; RNP (black), RNP and ssODN (red), pLV-ω1X6 virions (green), pLV-ω1X6 virions and ssODN (blue) compared to controls. (C–D) Loss of RNase activity as assessed by hydrolysis of yeast RNA. Residual yeast following exposure to SEA, visualized after gel electrophoresis; DW (negative control), RNase A, control SEA (from wt LE), other negative control SEAs; mock, sgRNA only, cas9 only, heated-pLV-ω1X6 + ssODN and the Δω1-SEAs from RNP, RNP with ssODN, pLV-ω1X6 virions, and pLV-ω1X6 virions with ssODN. (D) Intact yeast RNA (nanograms) remaining following incubation with SEA (mean ±SD, n = 6). More RNA remained following incubation with Δω1-SEA in all groups, RNP, RNP and ssODN, pLV-ω1X6 virions, and pLV-ω1X6 virions and ssODN treated SEA) (blue) than in the WT SEA controls (p≤0.0001). Among the gene edited experimental groups, more RNA remained when donor template was introduced at the same time as RNP or pLV-ω1X6 virions (p≤0.01, n = 11 by ANOVA). Significant differences were not apparent among the WT SEA control groups.

DOI: https://doi.org/10.7554/eLife.41337.010

The following figure supplement is available for figure 3:

**Figure supplement 1.** Quantitative PCR to estimate efficiency of programmed mutagenesis.
DOI: https://doi.org/10.7554/eLife.41337.011

## Diminished ribonuclease activity in CRISPR/Cas9-mutated schistosome eggs

The ribonuclease activity of the ω1 glycoprotein in SEA is associated with the Th2-polarized immune response that underpins the appearance of schistosome egg granulomata (*Everts et al., 2012*;

*Steinfelder et al., 2009*; *Everts et al., 2009*). Ribonuclease activity of SEA from control and experimental groups on substrate yeast RNA was investigated following CRISPR/Cas9 programmed mutation of ω1 mediated by the RNP and the pseudotyped lentiviral approaches with or without ssODN. Intact yeast RNA was evident in the DNase-RNase free condition (negative control), indicating absence of RNase activity in the reagents (200 ng yeast RNA at the outset). There was near complete hydrolysis of yeast RNA following exposure to RNase A (positive control);~1.4 ng of RNA remained intact. Wild type SEA exhibited marked RNase activity against the yeast RNA;~70 ng RNA remained intact after one hour, corresponding to >60% digestion. Incubation of the RNA with Δω1-SEA (i.e. SEA from the gene edited eggs) from the experimental groups, RNP, RNP +ssODN, pLV-ω 1X6, and pLV-ω1X6 + ssODN, resulted in ~30% substrate digestion, with 124, 140, 135 and 153 ng of RNA remaining, respectively. All conditions for programmed genome editing resulted in less digestion of the yeast RNA than for wild type SEA (p≤0.0001) (*Figure 3C and D*). Moreover, the Δω1-SEA with programmed knock-in exhibited less RNase activity than Δω1-SEA prepared without the donor ssODN repair template (p≤0.01, n = 6 by one-way ANOVA).

## Depleting SEA of ϖ1 downregulated Th2 response

The ω1 ribonuclease alone is capable of conditioning human monocyte-derived dendritic cells to drive Th2 polarization (*Everts et al., 2012*) and enhanced CD11b[+] macrophage modulation of intracellular toll like receptor (TLR) signaling (*Ferguson et al., 2015*; *Ferguson et al., 2016*). Ribonuclease ω1 inhibited TLR-induced production of IL-1β and redirected the TLR signaling outcome toward an anti-inflammatory profile via the mannose receptor (MR) and dectin (*Everts et al., 2012*; *Zaccone et al., 2011*; *Ritter et al., 2010*). The human monocytic cell line, THP-1, and the Jurkat human CD4[+] T cell line were employed to investigate the interaction of antigen-presenting cells and T cells (*Qin, 2012*; *Fuentes et al., 2002*). At the outset, the THP-1 cells were differentiated to macrophages for 48 hr, and subsequently pulsed with SEA or Δω1–SEA for 48 hr. Thereafter, the Jurkat CD4[+] T cells were added to the wells, and the co-culture continued for 72 hr. Representative cytokines, including IL-4, IL-5, IL-13, IL-2, IL-6, IL-10, TNF-α and IFN-γ, were quantified in supernatants of the co-cultures (*Figure 4*). SEA from ω1-mutated eggs reduced levels of Th2 cytokines, including IL-4 and IL-5, in comparison to wild-type SEA (p≤0.01), and a trend toward less IL-13 production was also observed (*Figure 4*). Reduced levels of IL-6 and TNF-α were also observed (p≤0.01, n = 4 by one-way ANOVA). By contrast, significant differences in IL-10 and IL-2 were not evident between the WT- and mutant-SEA groups. IFN-γ was not detected following pulsing with the WT-SEA or mutant-SEA (*Figure 4—figure supplement 1*).

## Granulomatous inflammation markedly reduced in lungs of mice injected with Δϖ1 eggs

Following the entrapment of eggs in the intestines, liver and eventually lungs, the glycosylated ω1 ribonuclease represents the principal stimulus that provokes the development of the circumoval granuloma, necessary for extravasation of the eggs (*Doenhoff et al., 1986*). A long-established model of the schistosome egg granuloma employs tail vein injection of eggs into mice, which leads to formation of circumoval granuloma in the mouse lung (*Boros and Warren, 1970*; *Eltoum et al., 1995*; *Wynn et al., 1993*). The latter approach has been extensively employed for immunopathogenesis-related studies of ω1 (*Hagen et al., 2014*). Accordingly, to elicit circumoval granulomas, ~3000 WT or Δω1 LE were injected into the lateral vein of the tail of BALB/c mice. The mice were euthanized 10 days after injection, and the entire left lung was removed, fixed, sectioned, and stained for histological analysis (*Figure 5*). Representative digital microscopic images of the whole mouse lungs acquired through high-resolution 2D digital scans are presented in *Figure 5*, panels A-G. At low magnification (2×), much more severe and widespread inflammation was visible in lungs exposed to WT eggs compared to Δω1-eggs. In addition, markedly more intense and dense focal inflammation was induced by WT compared to Δω1-eggs (*Figure 5B*). Granulomas were not seen in control naive mice not exposed to schistosome eggs (*Figure 5C*). At 20× magnification, vast disparity in volume of the circumoval granulomas was observed for WT versus Δω1 LE (*Figure 5A1–A2, D,E* and *Figure 5B1–B2, F and G*). The volume of granulomas surrounding single schistosome eggs was quantified; those surrounding WT eggs in lungs of the mice were 18-fold greater than for Δω1-eggs, $21 \times 10^{-2} \pm 1.61 \times 10^{-3}$ mm$^3$ and $0.34 \times 10^{-2} \pm 0.12 \times 10^{-4}$ mm$^3$ (mean ±S.E., 17–26

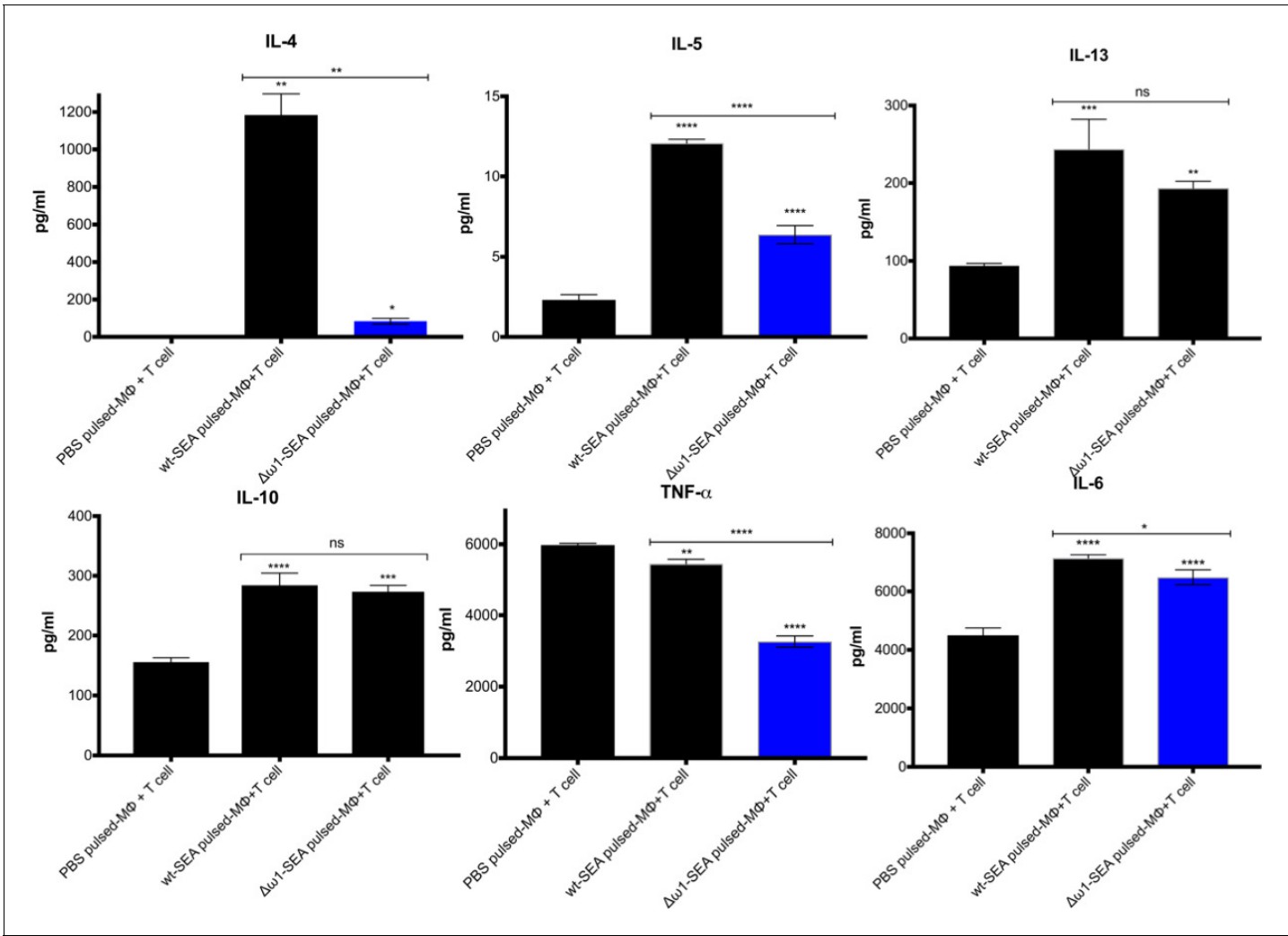

**Figure 4.** Reduced Th2 cytokine levels following exposure to Δω1-SEA. Reduction in Th2 cytokines IL-4 and IL-5 but not IL-13 following pulsing of Mφ (PMA induced-THP-1 cells) with Δω1-SEA prior to co-culture with human CD4[+] T cells (Jurkat cell line) compared with WT-SEA pulsed-Mφ (top panels). In addition, levels of IL-6 and TNF-α were reduced where Mφ were first pulsed with Δω1-SEA but not WT SEA. Differences were not evident for IL-10. The assay was carried out in triplicate; p<0.0001,≤0.0001, 0.0038 and 0.0252 indicated as ****, ***, ** and *, respectively (one-way ANOVA, multiple comparison, *n* = 4).

DOI: https://doi.org/10.7554/eLife.41337.012

The following figure supplement is available for figure 4:

**Figure supplement 1.** Levels of IL-2, IL-10 and IFN-γ following exposure to Δω1-SEA.

DOI: https://doi.org/10.7554/eLife.41337.013

granulomas per mouse), respectively (p<0.0001, n = 103–130 by Welch's *t*-test) (*Figure 5H*). The experiment was repeated with 3–4 mice per group with a similar outcome. The findings documented marked deficiency in the induction of pulmonary granulomas by the Δω1 compared to WT eggs of *S. mansoni*.

## Discussion

This report, and the accompanying article on the liver fluke *Opisthorchis viverrini* (*Arunsan et al., 2019*) pioneer programmed genome editing using CRISPR/Cas9 of trematodes and indeed genome editing for species of the phylum Platyhelminthes. Using *S. mansoni* as an exemplar, here we have demonstrated the activity and feasibility of gene knockout and knock-in in schistosomes. Programmed on-target editing was evidenced by site-specific mutations at the ω1 locus on

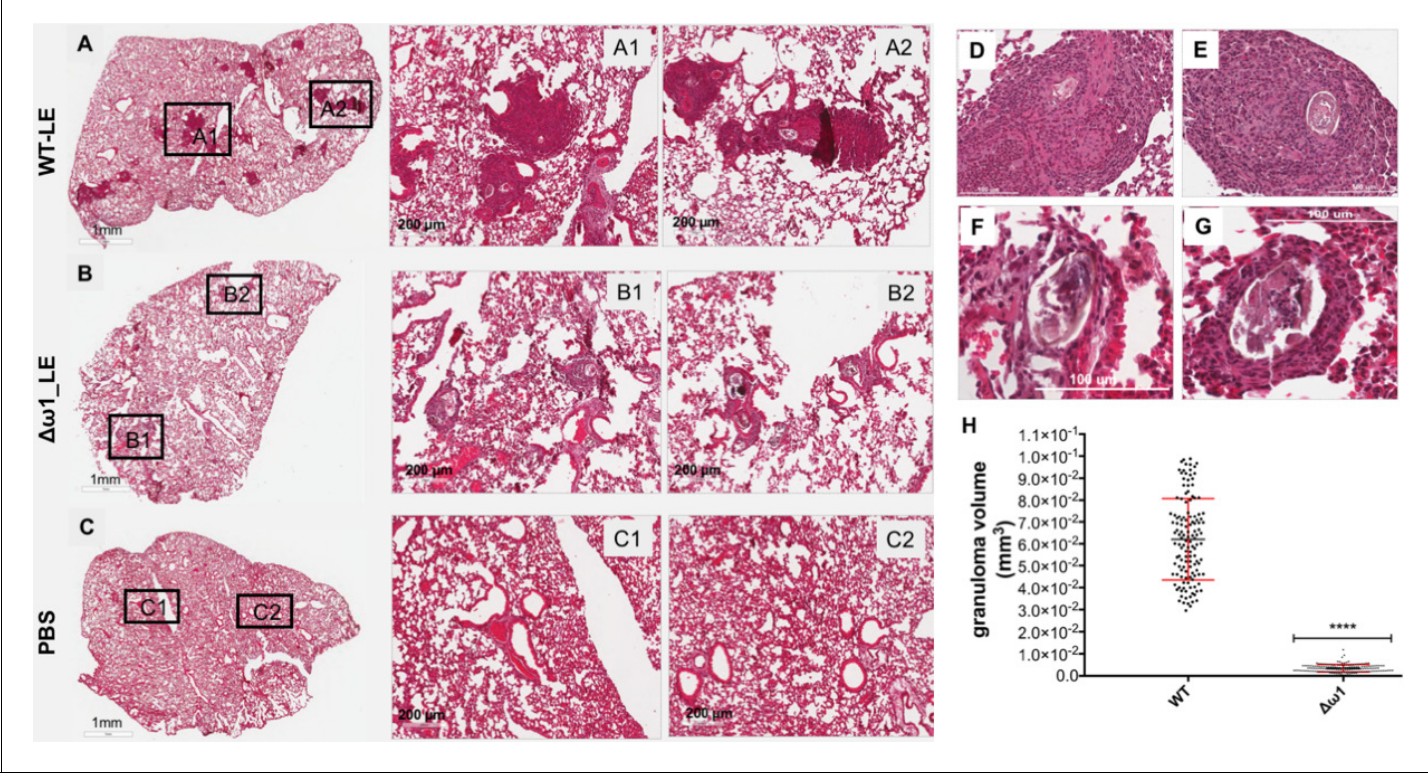

**Figure 5.** Pulmonary circumoval granulomas revealed attenuated granulomatous response to Δω1 schistosome eggs. Schistosome eggs (~3,000 eggs) that had been transduced with lentivirus virions encoding ω1-specific sgRNA and Cas9 in tandem with ssODN were introduced via the tail vein into mice. The mice were euthanized 10 days later; thin sections of the left lung were stained with H&E, and circumoval granulomas counted and measured. (**A**) Representative 2D scanned micrographs of granulomas inoculated with WT eggs (2× magnification) and 20× magnification (A1 and A2), and with Δω1 eggs; (**B**) (2×), B1 and B2 (20×). (**C**) control mouse lung. (**D**) and (**E**) Representative micrograph of individual, control eggs induced-granuloma that was counted to assess for granuloma volume. (**F**) and (**G**) Representative micrographs showing Δω1 egg induced-granulomas. All single egg induced-granuloma from WT and Δω1 eggs were measured and granuloma volume estimated. (**H**) Scatter plots of the volume (mm³) for individual granuloma, mean ± SE (red) are shown. The volumes of granulomas induced by Δω1 eggs were significantly smaller than those surrounding WT eggs (Welch's $t$-test, p≤0.0001, $n > 100$).

DOI: https://doi.org/10.7554/eLife.41337.014

chromosome 1. The chromosomal lesion was repaired by the non-homologous end joining (NHEJ) pathway (*Lieber, 2010*) in the absence of a donor oligonucleotide and by homology directed repair (HDR) when a single-stranded oligonucleotide donor template was provided (*Paquet et al., 2016*; *Zhang and Matlashewski, 2015*; *Yoshimi et al., 2016*).

To investigate the feasibility of genome editing, schistosome eggs were transfected with ribonucleoprotein particle (RNP) complexes and with lentiviral virions carrying the CRISPR/Cas9 components, in similar fashion to earlier reports in cell lines, tissues and entire organisms (*Kosicki et al., 2017*; *Holmgaard et al., 2017*; *Yu et al., 2018*; *Luo et al., 2018*). Delivery by RNPs facilitates immediate editing, although the short half-life of the RNP components may be disadvantageous. Delivery by plasmids and by viral-mediated infection may provide sustained Cas9 activity, transgene integration in non-dividing cells, and other advantages (*Hsu et al., 2014*; *Shalem et al., 2015*). Transfection by LV provides a hands-free approach to enable scaling of gene editing and the potential that less accessible and/or differentiated cells can be reached.

To examine the efficiency of programmed genome editing, several parallel approaches were undertaken including NGS- and quantitative PCR-based analysis of pooled genomic DNAs, analysis of levels of ω1-transcripts and ribonuclease, and immuno-phenotypic status of cultured cells and of mice exposed to gene edited eggs. Analysis of the deep-coverage nucleotide sequence reads of amplicons spanning the predicted DSB site in the ω1 locus revealed that ~4.5% of the reads were mutated by insertions, deletions and substitutions. The target locus was mutated by knock-in (KI) of

a ssODN repair template bearing short homology arms to mediate homology-directed repair following DSB at ω1, with an efficiency for HDR of 0.19% insertion of the donor transgene. Numerous substitutions in addition to deletions and insertions were seen, some of which may represent single nucleotide polymorphisms (SNPs) among the gene copies rather than genome editing-induced changes. As well as the anticipated ease of access by the RNPs, virions and the donor template ssODN, to cells proximal to the eggshell compared to the cells deeper within the egg, other factors also may have contributed to unevenness of CRISPR efficiency among the eggs and the cells within individual eggs. Given the presence of multiple copies of ω1 in the genome, the diverse organs, tissues and cells comprising the mature egg (*Jurberg et al., 2009*), the spectrum of development of the eggs in LE, and other factors, a mosaic of mutations would be expected. Some alleles might display HDR but not NHEJ, some NHEJ but not HDR, others both NHEJ and HDR, and others retain the wild-type genotype. HDR proceeds at cell division, with the cell at late S or G2 phase following DNA replication where the sister chromatid serves as the repair template; otherwise, NHEJ proceeds to repair the DSB (*Heyer et al., 2010*).

Sequence analysis of individual schistosomes rather than pools of the parasites would provide more information, including transfection/transduction efficiency and parasite-to-parasite mutation rates. However, the analysis would be constrained by the number of parasites that could be investigated at the genome level in terms of effort necessary for the numerous NGS groups in the computational analysis. Here, many thousands of individual eggs were simultaneously subjected to gene editing in order to provide sufficient quantities of RNA and gDNA for the downstream transcript level and NGS-based gene editing analyses. Although it is technically feasible to consider analysis of individual schistosome eggs, it would be challenging to reliably recover enough nucleic acids for the downstream analysis. Nonetheless, genotyping individual eggs or other developmental stages of the schistosome would be informative and straightforward to confirm the presence of transgenes following HDR of donor templates. Moreover, droplet digital PCR (ddPCR)-based analysis, a more sensitive approach than used here, is expected to provide a more sensitive and reliable quantification of gene-editing mutations. The ddPCR approach provides simultaneous assessment of both HDR and NHEJ, the repair pathways that resolve Cas9-catalyzed, double-stranded breaks, and also enable investigation of multiple, simultaneous editing conditions at the target locus (*Dibitetto et al., 2018*; *Miyaoka et al., 2018*). Newer technologies including single-cell genome sequencing will be able to precisely define not only individual target cells that were mutated, but also identify which kind of mutations arose in these individual cells (*Gawad et al., 2016*).

Expression levels of ω1 were diminished by as much as 83% relative to controls suggesting that Cas9 catalyzed the mutation of ω1, and that programmed Cas9-induced DSBs had been resolved by NHEJ and/or HDR. Knock-in of the ssODN repair template induced 81–83% reduction in ω1-specific mRNA levels, whereas downregulation of 45% to 67% followed the exposure of eggs to RNP or lentivirus without ssODN. Ribonuclease activity in ω1-mutated eggs was likewise significantly diminished. Curiously, less than 5% efficiency (NGS findings) in gene editing appeared to account for this markedly reduced (>80%) gene expression. The possibility of large-scale deletions, as reported in *Strongyloides stercoralis* (*Gang et al., 2017*), may provide an explanation for this apparent paradox. However, analysis of copy number by qPCR failed to reveal apparent differences for copy numbers of ω1 among treatment and control groups of eggs. An alternative explanation may be the tight, stage- and tissue-specific expression of ω1. Within the mature egg, the fully developed miracidium is surrounded by a squamous, syncytial epithelium termed variously as the envelope, the inner envelope, or von Lichtenberg's envelope (*Ashton et al., 2001*; *Mathieson and Wilson, 2010*; *Jurberg et al., 2009*; *Neill et al., 1988*). The inner envelope is metabolically active (*Mathieson and Wilson, 2010*) and is considered to be the site of synthesis of the ω1 T2 ribonuclease that is released from the egg into the granuloma (*Ashton et al., 2001*; *Mathieson and Wilson, 2010*; *Fitzsimmons et al., 2005*), along with other secreted/excreted proteins that facilitate egress of the egg from the venule and through the wall of the intestine (*Mathieson and Wilson, 2010*). Expression of ω1 is tightly, developmentally regulated: based on comparison of expression in the egg to expression in the miracidium, immunostaining and transmission electron microscopy of the mature egg, and on meta-analysis of transcriptomics sequence reads, all or most expression of ω1 occurs solely in the mature egg. Expression does not occur earlier in the embryo or immature egg or other developmental stages (*Everts et al., 2009*; *Ashton et al., 2001*; *Mathieson and Wilson, 2010*; *Fitzsimmons et al., 2005*; *Lu et al., 2018*; *Schramm et al., 2009*) (*Figure 1—figure supplement*

1C). The virions may have transduced only a small number of the cells within the mature egg because, following entry into the egg, the virus would be expected to first contact the inner envelope, rather than traveling further to the cells of the miracidium. Accordingly, the efficiency of gene editing may have ranged from high to low – from the exterior of the egg to its center through the cells of the egg due to ease of virus access from the culture supernatant, and displayed highest efficiency in the envelope where ω1 is expressed. In any case, this apparent paradox deserves deeper inquiry, for example by confirming the site of expression of ω1 by immunolocalization in the inner envelope of the fully mature egg.

Following mating of the adult female and male *S. mansoni*, schistosome oviposition commences at ~35 days after infection of the mouse. Thereafter the female schistosome continuously releases several hundred eggs each day. When released into the mesenteric veins from the adult female, the schistosome ovum contains a single-celled zygote surrounded by ~40 vitelline cells of maternal origin (*Mathieson and Wilson, 2010*). By six days later, the miracidium has developed within the eggshell into a multi-cellular, mobile, ciliated larva composed of organs, tissues, muscles and nerves (*Ashton et al., 2001*; *Jurberg et al., 2009*; *Neill et al., 1988*; *Mann et al., 2011*). Productive infection and transmission of schistosomiasis require the translocation of the egg across the wall of the bowel. However, many eggs fail to escape from the venules and are transported by the portal circulation to the sinusoids of the liver where they become entrapped. Maturation of the egg from the single-cell zygote within the eggshell at the time of oviposition to the fully mature miracidium within the eggshell takes about 7 days in vitro (*Rinaldi et al., 2012a*; *Jurberg et al., 2009*) and presumably a similar duration in vivo. Here, at necropsy of the mice, we anticipated that the LE preparation of eggs would include a spectrum of embryonic development of the schistosome egg from the blastula stage to mature miracidium containing organs, tissues and nerves. The fully formed egg is laid still undeveloped, without any cleavage. Once released from the female schistosome, eight embryonic stages can be defined. Stage one refers to early cleavages and the beginning of yolk fusion, whereas stage eight refers to the fully formed larva, presenting muscular contraction, cilia, and flame-cell beating (*Jurberg et al., 2009*). The envelope of the egg, the site of expression of ω1 (28, 30) first appears at stage three and is developed and secretory at the later stages (*Jurberg et al., 2009*).

To address the effects on the hallmark Th2 polarization of the immune response to schistosomiasis (*MacDonald et al., 2002*; *MacDonald and Pearce, 2002*), co-cultures of human macrophages and T cells were exposed to Δω1-SEA and mice were exposed to Δω1-eggs. Whereas wild type SEA polarized Th2 cytokine responses including IL-4 and IL-5 in the co-cultures, significantly reduced levels of these cytokines were observed after exposure to ω1-mutated SEA. Moreover, following introduction of eggs into the tail vein of mice, the volume of pulmonary circumoval granulomas around Δω1 eggs was enormously reduced compared to those provoked by wild-type eggs. Although this outcome extends earlier findings using lentiviral transduction of eggs of *S. mansoni* to deliver micro-RNA-adapted short hairpin RNAs aiming to silence expression of ω1 (*Hagen et al., 2014*), the decrement in granuloma volume was vastly more marked in the present study. In addition to incomplete disruption of all copies of ω1, residual granulomas containing mutant eggs may be due to the presence of other Th2-polarizing components within SEA (*Kaisar et al., 2018*). Given that the T2 ribonuclease ω1 is the major type 2-polarizing protein among egg-secreted proteins (*Steinfelder et al., 2009*; *Everts et al., 2009*), these findings of a phenotype characterized by the absence of or diminutive pulmonary granulomas provide functional genomics support for this earlier advance (*Everts et al., 2012*). Nonetheless, given that hepatointestinal disease is characteristic of the infection with *S. mansoni*, changes in granuloma formation in the liver and intestines triggered by mutant egg will be prioritized in future studies.

This study provides a blueprint for editing other schistosome genes and those of parasitic platyhelminths at large. However, hurdles remain. We suggest the following (non-inclusive) list of near term priorities with respect to development of programmed gene editing for functional genomics in flatworms: the delivery of programmed genome editing to the germ line, including transgenes that confer constitutive or conditional expression of Cas9 (*Idoko-Akoh et al., 2018*; *Chaverra-Rodriguez et al., 2018*); HDR-catalyzed insertion of surrogate reporter, antibiotic resistance markers (*Yan et al., 2018*) or antimetabolites to facilitate drug selection of the mutants (*Yoshimi et al., 2016*; *Wu et al., 2014*; *Yan and Finnigan, 2018*); and dual guide systems (*Teixeira et al., 2018b*; *Teixeira et al., 2018a*; *Gong et al., 2017*). Focusing on schistosomes, establishment of lines of transgenic *S. mansoni* by retroviral-based transgene integration into the zygote within the newly laid

egg has been described (*Rinaldi et al., 2012a*). Beyond the zygote, the germ line is also comparatively more accessible at several developmental stages, including the daughter sporocysts (*Wang et al., 2013*). Concerning somatic tissue or whole worm gene editing approaches, synchronizing the miracidial development in the cultured egg is straightforward by maintaining LE in vitro for 7 days. In this way, the reproducibility of somatic editing might be improved, as mosaicism outcomes would be reduced since each mature egg enclosed the fully developed miracidium with the full complement of cells (*Jurberg et al., 2009*). For HDR, jointly supplying both anti-sense and sense long single-stranded donor DNAs in trans should increase the likelihood of obtaining mutant homozygotes rather than heterozygotes (*Paquet et al., 2016*; *Gantz and Bier, 2015*). In turn, HDR focused programmed gene editing to generate bi-allelic mutation would hasten the heritable spread of transgenes to F2 and beyond.

Gene editing the germ line followed by establishment of mutant schistosomes represents a powerful strategy to determine the role and potential fitness cost that result from deletion of the ω1 gene. However, given the established role of the ω1 T2 ribonuclease in provoking the circumoval granuloma and given the predicted role of the granuloma in translocation of the schistosome egg from the intestinal wall to the lumen of the bowel, a line of Δω1 schistosomes may not be transmissible via eggs shed in the host feces. Nonetheless, a line of Δω1 *S. mansoni* schistosomes might be rescued by experimentally transferring Δω1 eggs from livers of mice into water, for continuation of the developmental cycle. If so, not only would this definitively demonstrate both the pathophysiological role of ω1 but also would demonstrate essentiality of ω1 for successful parasitism. Mutant parasite lines can be predicted to enable more comprehensive understanding of the pathobiology of these neglected tropical disease pathogens and facilitate access to novel insights and strategies for disease management.

To conclude, these findings reported here confirmed that somatic genome editing of schistosome eggs led to functional knockout of the ω1 T2 ribonuclease, and revealed the likely presence of genetic mosaicism in mutant cells resulting from gene editing (*Mehravar et al., 2018*). The genome-edited eggs exhibited loss of function of ω1 but remained viable. Programmed mutation of ω1 using CRISPR/Cas9 not only achieved the aim of establishing the applicability of genome editing for functional genomics of schistosomes but also demonstrated manipulation of a gene expressed in the schistosome egg, the developmental stage central to the pathophysiology of schistosomiasis.

## Materials and methods

**Key resources table**

| Reagent type (species) or resource | Designation | Source or reference | Identifiers |
|---|---|---|---|
| Parasite | *S. mansoni* eggs | BEI resources | |
| Synthetic RNA | Guide RNA targeting exon 6 of omega-1 | Thermo Fisher Scientific | Custom |
| Plasmid construct | Lentiviral CRISPR Cas9 construct containing human U6 promoter driving gRNA and human EF1 promoter driving Cas9 | Sigma | All-in-one vector (Custom) |
| Oligonucleotides | single strand oligodeoxynucleotide | Eurofin Genomics | Custom |
| *Streptococcus pyogenes*, nuclease | Cas9 nuclease | Dharmacon | CAS11201 |
| *Homo sapiens*, cell line | HEK293 | ATCC | CRL-1573, RRID:CVCL_0045 |
| *Homo sapiens*, cell line | THP-1 | ATCC | TIB-202, RRID:CVCL_0006 |
| *Homo sapiens*, cell line | Jurkat | ATCC | CRL-2901, RRID:CVCL_U620 |

*Continued on next page*

*Continued*

| Reagent type (species) or resource | Designation | Source or reference | Identifiers |
|---|---|---|---|
| Commercial kit | Lentiviral Packaging Mix | Sigma-Aldrich | SHP001 |
| Commercial reagent | Lentiviral concentrator | Takara Bio | 631231 |
| Commercial kit | Lentiviral quantification | Takara Bio | 631280 |
| Commercial kit | QIAseq 1-step Amplicon Library kit | Qiagen | 180412 |
| Commercial reagent | GeneRead Adaptor I | Qiagen | 180986 |
| Commercial kit | QIAseq Library Quant System | Qiagen | QSIL-003 |
| Commercial kit | Th1/Th2/Th17 ELISA multi-analyte test kit | Qiagen | 336161 |

## Ethics statement

Mice experimentally infected with *S. mansoni*, obtained from the Biomedical Research Institute (BRI), Rockville, MD were housed at the Animal Research Facility of the George Washington University Medical School, which is accredited by the American Association for Accreditation of Laboratory Animal Care (AAALAC no. 000347) and has an Animal Welfare Assurance on file with the National Institutes of Health, Office of Laboratory Animal Welfare, OLAW assurance number A3205-01. All procedures employed were consistent with the Guide for the Care and Use of Laboratory Animals. The Institutional Animal Care and Use Committee (IACUC) of the George Washington University approved the protocol used for maintenance of mice and recovery of schistosomes. Studies with BALB/c mice involving tail vein injection of schistosome eggs and subsequent euthanasia using over-dose of sodium pentobarbital was approved by the IACUC of BRI, protocol 18–04, AAALAC no. 000779 and OLAW no. A3080-01.

Mice were euthanized 7 weeks after infection with *S. mansoni*, livers were removed at necropsy, and schistosome eggs recovered from the livers, as described (*Dalton et al., 1997*). The liver eggs termed 'LE' were maintained in DMEM medium supplemented with 10% heat-inactivated fetal bovine serum (FBS), 2% streptomycin/penicillin at 37°C under 5% $CO_2$ in air for 18–24 hr, after which LE was exposed to electroporation with RNP or transduction by LV for programmed gene editing (*Mann et al., 2010*; *Mann et al., 2014*). Polymyxin B (10 µg/ml) was added to the cultures twice daily to neutralize lipopolysaccharide (LPS) (*Cardoso et al., 2007*). Soluble egg antigen (SEA) was prepared from these eggs, as described (*Dunne et al., 1991*; *Boros and Warren, 1970*). In brief, the homogenate of eggs in 1× PBS containing protease inhibitor cocktail (Sigma) was frozen and thawed twice, clarified by centrifugation at 13,000 rpm, 15 min, 4°C, the supernatant passed through a 0.22 µm pore size membrane. Protein concentration of the supernatant (SEA) was determined by the Bradford Protein Assay (*Bradford, 1976*) and aliquots of the SEA stored at −80°C.

## Guide RNAs, Cas9, and single-stranded DNA repair template

Single guide RNA (sgRNA) was designed using the web-based tools at http://bioinfogp.cnb.csic.es/tools/breakingcas/ (*Oliveros et al., 2016*) to predict cleavage sites for the *Streptococcus pyogenes* Cas9 nuclease within the genome of *S. mansoni*. The sgRNA targeted exon 6 of the ω1 gene, Smp_193860, www.genedb.org, residues 3,808–3,827, adjacent to the protospacer adjacent motif, AGG (*Figure 1A*). This is a multi-copy gene with at least five copies of ω1 located in tandem on chromosome 1 (*Protasio et al., 2012*). To infer the gene structure of Smp_193860 in the *S. mansoni* V5 genome assembly more accurately, the omega-1 mRNA DQ013207.1 sequenced by *Fitzsimmons et al. (2005)* was used to predict the gene structure with the exonerate software, by aligning it to the assembly using the exonerate options '–model coding2genome' and '–maxintron 1500'. The Smp_193860 copy of ω1 includes nine exons interspersed with eight introns (6196 nt) (*Figure 1A*).

Synthetic gRNA (sgRNA), ω1-sgRNA was purchased from Thermo Fisher Scientific (Waltham, MA). A double-stranded DNA sequence complementary to the sgRNA was inserted into lentiviral gene editing vector, pLV-U6g-EPCG (Sigma), which encodes Cas9 from *S. pyogenes* driven by the

eukaryotic (human) translation elongation factor 1 alpha 1 (tEF1) promoter and the sgRNA driven by the human U6 promoter (*Figure 1C*). The pLV-U6g-EPCG vector is tri-cistronic and encodes the reporter genes encoding puroR and GFP, in addition to Cas9 (*Fitzsimmons et al., 2005*). This gene-editing construct, targeting exon 6 of ω1 Smp_193860, was termed pLV-ω1X6. A single-stranded oligodeoxynucleotide (ssODN) (*Lok et al., 2017*), which included homology arms of 50 nt each in length at the 3' (position 3775–3824 nt) and 5' (3825–3874 nt) flanks and a small transgene (5'-TAAGTGACTAGGTAACTGAGTAGC-3', encoding stop codons (six) in all open-reading frames) (*Figure 1B*), was synthesized by Eurofin Genomics (Louisville, KY). An oligonucelotide primer that included this sequence was employed in PCRs to investigate the presence of CRISPR/Cas9-programmed insertion of the transgene (*Supplementary file 2*).

## Transfection of schistosome eggs with a Cas9/guide RNA complex

For the ribonucleoprotein (RNP) complex of the ω1-sgRNA and recombinant Cas9 from *Streptococcus pyogenes*, 6 µg of ω1-sgRNA and 6 µg of Cas9-NLS nuclease (Dharmacon, Lafayette, CO) were mixed in 100 µl Opti-MEM (Sigma) to provide 1:1 ratio w/w RNP. The mixture was incubated at room temperature for 10 min, pipetted into a 4 mm pre-chilled electroporation cuvette containing ~10,000 LE in ~150 µl Opti-MEM, subjected to square wave electroporation (one pulse of 125 volts, 20 ms) (BTX ElectroSquarePorator, ECM830, San Diego, CA). The electroporated eggs were incubated for 5 min at room temperature, and maintained at 37°C, 5% $CO_2$ in air for 3, 5 and 7 days. To investigate whether homology-directed repair (HDR) could catalyze the insertion of a donor repair template, 6 µg ssODN was mixed with RNP and the LE before electroporation. In a second approach (above), the ssODN was delivered to LE by electroporation at ~24 hr after the lentiviral transduction of the LE. The eggs were collected 3, 5 and 7 days later and genomic DNA recovered from LE. The negative controls included LE subjected to electroporation in the presence of only Opti-MEM, only Cas 9, only sgRNA, and only ssODN.

## Transduction of schistosome eggs with lentiviral particles

*Escherichia coli* Zymo 5α (Zymo Research) cells were transformed with lentiviral plasmid pLV-ω1X6 and cultured in LB broth in 100 µg/ml ampicillin at 37°C, agitated at 225 rpm for ~18 hr, after which plasmid DNA was recovered (GenElute Plasmid purification kit, Invitrogen). A lentiviral (LV) packaging kit (MISSION, Sigma-Aldrich) was used to prepare LV particles in producer cells (human 293T cell line). In brief, $3.5 \times 10^5$ of 293T cells/well were seeded in a six-well tissue culture plate in DMEM supplemented with 10% heat-inactivated fetal bovine serum (FBS), 2 mM L-glutamine, 1% penicillin/streptomycin and cultured at 37°C, 5% $CO_2$ for 18 hr. The producer cells were transfected using FUGENE HD (Promega) with pLV-ω1X6 and LV packaging mix containing two additional plasmids; one plasmid that expressed HIV structural and packaging genes and another that expressed the pseudotyping envelope protein Vesicular Stomatitis Virus Glycoprotein (VSVG). Subsequently, the transfection mixture (50 µl; 500 ng plasmid DNA, 4.6 µl packaging mix, 2.7 µl of FUGENE HD in Opti-MEM) was dispensed drop wise into each well on the plate. Sixteen hours later, the media were removed from the transfected cells, replaced with pre-warmed complete DMEM, and cells cultured for 24 hr. The supernatant, containing VSVG-pseudotyped LV particles was filtered through 22 µm pore size membranes (*Suttiprapa et al., 2016*), and stored at 4°C. Additional pre-warmed complete DMEM was added to the well, for culture for a further 24 hr. The supernatant was collected as above, combined with the first supernatant and concentrated (Lenti-X concentrator, Takara Bio, Mountain View, CA). Virion titer was estimated by two methods; first, by use of Lenti-X-GoStix (Takara Bio) to establish the presence of functional virions at $>10^5$ infectious units (IFU)/ml, and second, by reverse transcriptase assay (*Suttiprapa et al., 2016*; *Marozsan et al., 2004*) to quantify levels of active virions. Virions with counts of $\sim 4 \times 10^6$ count per minute (cpm)/ml were aliquoted and stored at −80°C.

To transduce LE with LV, ~10,000 eggs were incubated for 24 hr in complete DMEM containing 500 µl of $\sim 4 \times 10^6$ cpm/ml VSVG-LV virions. Thereafter, the LE were washed three times in 1× PBS and transfected with ssODN (6 µg) by square wave electroporation. The further steps (*Suttiprapa et al., 2012b*) with subsequent transfection with heat-inactivated pLV virions at 70°C for 4 hr with ssODN, transfection with ssODN in the absence of virions or Opti-MEM only served as negative controls.

## PCR amplification of diagnostic transgene to detect knock-in into exon 6 of ϖ1

For each DNA sample, four separate PCR assays using four distinct primer pairs (*Supplementary file 2*) were carried out. The first ω1 primer pair, to amplify locations 3,751–4,740 nt of Smp_193860, was employed as positive control for the presence of genomic DNA with the Smp_193860 copy of ω1. The other primer pairs shared one forward primer complementary to the knock-in 24 nt transgene with three reverse primers, Sm ω1-R1, -R2 and -R3 at positions 3,966–3,984, 4,066–4,085 and 4,102–4,121 nt, respectively, binding to three sites downstream of the ω1 predicted DSB site (*Figure 2A*, *Supplementary file 2*) (*Lok et al., 2017*). The PCR mix included 10 µl Green GoTaq DNA polymerase mix (Promega) with 200 nM of each primer and 10 ng genomic DNA. Thermal cycling conditions involved denaturation at 95℃, 3 min followed by 30 cycles of 94℃, 30 s, 60℃, 30 s and 72℃, 30 s and a final extension at 72℃ for 5 min. Following agarose gel electrophoresis (1.2% agarose/TAE), amplicons of the expected sizes were recovered from gels and ligated into pCR4-TOPO (Thermo Fisher). *E. coli* Zymo 5α competent cells were transformed with the ligation products, several colonies of each transformant were grown under ampicillin selection, plasmid DNA purified, and the inserts sequenced to confirm the presence and knock-in of the transgene (*Figure 1C*).

## Estimation of mutation efficiency

We used the egg genomic DNA templates directly for qPCR as described (*Yu et al., 2014*), with slight modification which enabled estimation of the efficiency of CRISPR-mediated mutagenesis at the target locus without the need to normalize the experimental and control template DNAs. The general approach makes use of the fact that the binding of a primer overlapping the sgRNA site was compromised in programmed mutagenized egg (LE) genome(s), resulting in delayed amplification, whereas binding of a flanking primer pair was unaffected. The 'OUT' (flanking) primer pair encompassed at least 50 bp surrounding the sgRNA binding region. The 'OVR' (overlapping) primer pair used one of the Smω1-OUT primers and another primer that bound the 20 bp of the sgRNA target sequence. The 3′ side of the Smω1-OVR primers bound immediately upstream of the NGG motif, as the majority of indels would be expected to affect positions −1 to −10 of the binding site (*Canver et al., 2018*).

At days 3, 5 and 7 following transfections with or without ssODN, genomic DNA was isolated from LE. Using 5 ng of DNA template, separate 20 µl OUT and OVR qPCR reactions were undertaken. Quantitative PCR was performed with SsoAdvanced SYBR Green Supermix (Bio-Rad, 172–5271) using a Bio-Rad iQ5 Real-Time PCR system, with qPCR conditions at initial 95℃ for 30s, 40 cycles, 95℃ for 10s, 60° C for20 s. Oligonucleotide primer sequences are provided in *Supplementary file 2*.

The ratio of the qPCR quantification cycle values for the control Smω1-OVR and control Smω1-OUT primers reflected the differences in amplification of the two primer pairs on control DNA template. This might have been due to inherent differences in amplification that exist even between perfectly complementary primer pairs. In contrast, the OVR/OUT ratio in mutant DNA reflected both this difference in amplification between the primer pairs and the loss of the OVR binding site due to CRISPR-introduced indels. A comparison of the OVR/OUT quantification cycle ratios of control versus mutated genomes thus reflected the efficiency of mutagenesis. The CRISPR efficiency was calculated by the Ct ratio of OVR:OUT, after which indel/substitution mutation percentage was estimated as follows:

% Relative fold amplification = 100 × (B/A)
A = Ct ratio of OVR:OUT from control group
B = Ct ratio of OVR:OUT from experiment group

## Illumina sequencing

Pooled LE DNA samples from six independent KI experiments of pLV-ω1X6 with ssODN were used as the template to amplify the on-target DNA fragment using MiSeq primers (*Figure 2A*) with High Fidelity *Taq* DNA polymerase (Thermo Fisher). PCR reactions were set up with 10 ng LE DNA samples from the KI experiment in 25 µl reaction mix using the HiFidelity Taq DNA polymerase (Thermo Fisher) following the PCR program 94℃ for 3 min of denaturation followed by 30 cycles of 94℃ for 30s, 60℃ or 54℃ for 30 s, 72℃ for 45s and final extension at 72℃ for 2 min. The expected size of

the amplicon flanking predicted DSB was 202 bp. Amplicons of this size were purified using the Agencourt AMPure XP system (Beckman Coulter). Amplicons generated from four different PCR reactions from each sample were pooled, and 100 ng of amplicons from each sample was used to construct the uniquely indexed paired-end read libraries using the QIAseq 1-step Amplicon Library Kit (Qiagen) and GeneRead Adapter I set A 12-plex (Qiagen). These libraries were pooled, and the library pool was quantified using the QIAseq Library Quant System (Qiagen).

Samples (*Supplementary file 3*) were multiplexed (10 samples) and each run on four MiSeq lanes. After sequencing, the fastq files for each particular sample were merged. Samples 1–6, 8 and 10 were prepared using an annealing temperature of 54°C. Samples 7 and 9 were prepared using an annealing temperature of 60°C, and included an extra 10 bp at the start of the MiSeq sequences, 'GTTTTAGGTC', present upstream of the 5' primer in the genomic DNA. We trimmed this sequence from the reads using cutadapt v1.13 (*Martin, 2011*). To detect HDR events, computational software program CRISPResso (http://crispresso.rocks/) (*Pinello et al., 2016*; *Canver et al., 2018*) was employed using a window size of 500 bp (-w 500) with the reference amplicon according to gene Smp_193860 in the *S. mansoni* V7 assembly, and with the –exclude_bp_from_left 25 and –exclude_bp_from_right 25 options in order to disregard the (24 bp) primer regions on each end of the amplicon when indels are being quantified. A window size of 500 nt was employed to include the entire amplicon. In order to search for HDR events, CRISPResso checked for HDR events (using –e and –d options) in treatment groups including controls. To infer frameshifts using CRISPResso the –c option was used, giving CRISPResso the coding sequence from positions 42–179 of the amplicon. To confirm the insertions of the knock-in sequences reported by CRISPResso (right side column in *Supplementary file 3*), we took all insertions of 20–28 bp reported by CRISPResso, and calculated their percent identity to the expected knock-in sequence using ggsearch v36.3.5e in the fasta package (*Pearson and Lipman, 1988*). An insertion was considered confirmed if it shared ≥75% identity to the expected donor knock-in sequence.

## GenBank/EMBL/DDBJ
Database accessions
Sequence reads from the amplicon NGS libraries are available at the European Nucleotide Archive, study accession number ERP110149. Additional information is available at Bioproject PRJNA415471, https://www.ncbi.nlm.nih.gov/bioproject/PRJNA415471 and GenBank accessions SRR6374209, SRR6374210.

## Copy number estimation for ϖ1
A quantitative PCR to estimate the relative copy number of ω1 was performed using Kapa SYBR FAST Universal qPCR mastermix (KK4602) on 1 ng of gDNA templates (pooled genomic DNA samples from six biological replicates) isolated from control and test samples, in 20 μl volumes. A primer pair of OMGgRNA1F and OMGgRNA1R was used to amplify the ω1 gRNA target region and SmGAPDH (*Supplementary file 2*) as a reference single-copy gene (primers shown in *Supplementary file 2*). The PCR efficiencies for primer pairs were estimated by titration analysis to be 100% ± 5 (*Ginzinger, 2002*) and qPCRs were performed in triplicate in 96-well plates, with a denaturation step at 95°C of 3 min followed by 40 cycles of 30 s at 95°C and 30 s at 55°C, in thermal cycler fitted with a real time detector (StepOnePlus, Applied Biosystems). The relative quantification assay $2^{-\Delta\Delta Ct}$ method (*Livak and Schmittgen, 2001*) was used to ascertain the relative copy number of ω1. Relative copy number of ω1 in the CRISPR/Cas9 treated groups reflects the fold change of ω1 copy number normalized to the reference gene (SmGAPDH-F and -R primers) and relative to the untreated control group (calibrator sample with relative ω1 copy number = 1).

## Gene expression for ϖ1 mRNA
Total RNAs from schistosome eggs were extracted using the RNAzol RT reagent (Molecular Research Center, Inc), which eliminates contaminating DNA (*Chomczynski, 1993*), and concentration and purity determined using a spectrophotometer ($OD_{260/280}$ ~2.0). Reverse transcription (RT) of the RNA (500 ng) was performed using iScript Reverse Transcript (Bio-Rad), after which first strand cDNA was employed as template for qPCRs using SsoAdvanced Universal SYBR Green Supermix (Bio-Rad) performed in triplicates in an iQ5 real-time thermal cycler (Bio-Rad). RT-qPCR reaction

mixtures included 2 μl first strand cDNA, 5 μl SsoAdvanced Universal SYBR Green Supermix, and 300 nM schistosome gene-specific primers. *Supplementary file 2* provides details of the oligonucleotide primers. Thermal cycling included denaturation at 95°C for 30 s, 40 amplification cycles each consisting of denaturation at 95°C for 15 s and annealing/extension at 60°C for 30 s, and a final melting curve. The output was analyzed using the iQ5 software (BioRad). Relative expression was calculated using the $2^{-\Delta\Delta Ct}$ method and normalized to schistosome GAPDH expression (*Livak and Schmittgen, 2001*); data are presented as transcript levels (three replicates) compared to the wild-type LE (100%), and fold change reported as mean relative expression ±SD ($n$ = 11).

## Nuclease activity of ϖ1

A stock solution of yeast RNA (Omega Bio-tek, Norcross, GA) was prepared at 1.0 μg/μl, 50 mM Tris-HCl, 50 mM NaCl, pH 7.0. Yeast RNA (200 ng) was incubated with 2 μg SEA from control and experimental groups individually at 37°C for 60 min. SEA investigated here, named Δω1-SEA, was extracted from LE transduced with pLV- ω1X6 virions and ssODN, pooled from six biological replicates. RNase A, an endoribonuclease from bovine pancreas (Thermo Fisher) served as a positive control enzyme, whereas yeast RNA in reaction buffer only served as the negative control. The RNase activity of ω1 in wild-type SEA or Δω1-SEA was analyzed by visualizing and quantifying the substrate that remained following enzymolysis by agarose gel electrophoresis and staining with ethidium bromide. The yeast RNA digestion by control SEAs or Δω1-SEA were set up in triplicates, with quantity of residual RNA determined by densitometry (*Ke et al., 2017*).

## Macrophage polarization by WT or Δϖ1-SEA and T-cell activation in vitro

Human monocytic THP-1 cells were maintained in Roswell Park Memorial Institute medium (RPMI) 1640 (Thermo Fisher Scientific) containing 10% (v/v) FBS with 4 mM L-glutamine, 25 mM HEPES, 2.5 g/L D-glucose at 37°C in 5% $CO_2$ in air. THP-1 cells were non-adherent cells. In a 6-well plate, THP-1 monocytes ($3 \times 10^5$ cells in each well) were differentiated into macrophages (Mϕ) by incubation in 150 nM phorbol 12-myristate 13-acetate (PMA) (Sigma) for 48 hr (*Genin et al., 2015*). Mϕ were exposed to SEA (50 ng/ml) or Δω1-SEA (50 ng/ml) (from LE transduced with pLV-ω1X6 virions and ssODN) for 48 hr. To investigate macrophage and T cell interactions, Mϕ cells were pulsed with 50 ng/ml SEA or Δω1-SEA and thereafter co-cultured in direct contact with Jurkat (human CD4[+] T) cells. Nine $\times 10^5$ Jurkat were added to Mϕ with direct contact and were co-cultured for an additional 72 hr. Cell-free supernatants from the co-cultures were collected to quantify secretion of T helper cell cytokines including IL-4, IL-5, IL-13, IL-10, TNF-α, IL-6, IL-2 and IFN-γ by enzyme linked immunosorbent assay (Multiplex Human Cytokine ELISA kit, Qiagen) (*Schmid and Varner, 2010*). The assay included positive controls for each analyte, which were provided in the kit. Three biological replicates were undertaken.

## Cell lines

Human HEK293 cells, Jurkat and THP-1 cell lines were obtained from ATCC. All three cell lines were authenticated by ATCC using STR profiling by PCR and were confirmed in our laboratory to be *Mycoplasma*-free using the Lookout Mycoplasma PCR detection kit (Sigma-Aldrich).

## Schistosome egg-induced primary pulmonary granulomas

For induction of circumoval, egg-induced granulomas in the lungs of mice, 8 weeks old female (*Ashton et al., 2001*) BALB/c mice were injected with 3,000 WT eggs or Δω1-eggs (from experiment pLV-ω1X6 with ssODN) or 1× PBS as negative control by tail vein, as described (*Wynn et al., 1993*). The mice were euthanized 10 days later. Each group included 3 or five mice, two biological replicates were undertaken, totaling 6–10 mice for each treatment group. Before starting the experiment, mice were allocated randomly to the control or experimental treatment groups. For histopathological assessment of granuloma formation, the left lung was removed at necropsy and fixed in 10% formalin in pH 7.4 buffered saline for 24 hr, after which it was dehydrated in 70% ethanol, and clarified in xylene. The fixed lung tissue was embedded in paraffin and sectioned at 4-μm-thickness by microtome. Thin sections of the left lung lobe were mounted on glass slides and fixed at 58–60°C. Subsequently, rehydrated sections were stained with hematoxylin-eosin (H&E) for

evaluation of inflammatory infiltrates and cellularity of granulomas. Digital images were captured using a 2D glass slide digital scanner (Aperio Slide Scanner, Leica Biosystems, Vista, CA) and examined at high magnification using the Aperio ImageScope software (Leica) (*Eltoum et al., 1995*; *Cheever et al., 1992*). The longest (R) and shortest (r) diameters of each granuloma containing a single egg were measured with an ocular micrometer, and the volume of the granuloma calculated assuming a prolate spheroidal shape, using $4/3\pi Rr^2$ (*Ashton et al., 2001*). All granulomas in all slides from the left lung of the mice, 15 slides per treatment group, were measured; in total, >100 granulomas from each treatment group.

## Biological and technical replicates, statistics

In the assays above, we performed two or more biological replicates. These biological replicates represented parallel measurements of biologically discrete samples in order to capture any random biological variation. Technical replicates were undertaken as well; these represented two or three repeated measurements of the same sample undertaken as independent measurements of the random noise associated with the investigator, equipment or protocol.

Means for experimental groups were compared to controls by one-way ANOVA and, where appropriate, by two-tailed Student's $t$-test and Welch's unequal variances $t$-test (GraphPad Prism, La Jolla, CA). Values for $p$ of $\leq 0.05$ were considered to be statistically significant.

## Acknowledgements

We thank Dragana Jankovic, Alan Sher, Thomas Wynn, Michael Bukrinsky, Larisa Dubrovsky, Arnon Jurberg, Meredith Brindley, Robert Thompson and Thiago De Almeida Pereira for advice and technical assistance. Schistosome-infected mice and snails were provided by the NIAID Schistosomiasis Resource Center of the Biomedical Research Institute, Rockville, Maryland through NIH-NIAID Contract HHSN272201000005I for distribution through BEI Resources.

## Additional information

### Funding

| Funder | Grant reference number | Author |
|---|---|---|
| National Institute of Allergy and Infectious Diseases | R21AI109532 | Gabriel Rinaldi |
| Thailand Research Fund | PHD/0011/2555 | Apisit Chaidee |
| Thailand Research Fund | PHD/0047/2556 | Pannathee Prangtaworn |
| Thailand Research Fund | PHD/00531/2556 | Rutchanee Rodpai |
| Wellcome | 107475/Z/15/Z | Karl F Hoffmann |
| Wellcome | WT 098051 | Matthew Berriman |
| MaxMind Inc | | Paul J Brindley |

The funders had no role in study design, data collection and interpretation, or the decision to submit the work for publication.

### Author contributions

Wannaporn Ittiprasert, Conceptualization, Resources, Data curation, Software, Formal analysis, Supervision, Validation, Investigation, Visualization, Methodology, Writing—original draft, Writing—review and editing; Victoria H Mann, Resources, Investigation, Visualization, Methodology, Project administration, Writing—review and editing; Shannon E Karinshak, Investigation, Visualization, Writing—review and editing; Avril Coghlan, Resources, Data curation, Software, Formal analysis, Validation, Investigation, Visualization, Methodology, Writing—original draft, Writing—review and editing; Gabriel Rinaldi, Resources, Formal analysis, Supervision, Validation, Investigation, Visualization, Methodology, Writing—original draft, Writing—review and editing; Geetha Sankaranarayanan, Validation, Investigation, Methodology, Writing—review and editing; Apisit Chaidee, Investigation,

Writing—review and editing; Toshihiko Tanno, Resources, Data curation, Software, Formal analysis, Validation, Investigation, Methodology, Writing—review and editing; Chutima Kumkhaek, Resources, Software, Investigation, Methodology, Writing—review and editing; Pannathee Prangtaworn, Christina J Cochran, Patrick Driguez, Rutchanee Rodpai, Investigation, Methodology, Writing—review and editing; Margaret M Mentink-Kane, Resources, Investigation, Visualization, Methodology, Writing—review and editing; Nancy Holroyd, Resources, Data curation, Validation, Investigation, Visualization, Methodology, Writing—review and editing; Alan Tracey, Resources, Data curation, Software, Formal analysis, Validation, Investigation, Visualization, Methodology, Writing—review and editing; Bart Everts, Resources, Investigation, Methodology, Writing—review and editing; Cornelis H Hokke, Resources, Validation, Investigation, Methodology, Writing—review and editing; Karl F Hoffmann, Conceptualization, Supervision, Funding acquisition, Project administration, Writing—review and editing; Matthew Berriman, Resources, Data curation, Software, Formal analysis, Supervision, Funding acquisition, Validation, Investigation, Methodology, Project administration, Writing—review and editing; Paul J Brindley, Conceptualization, Resources, Data curation, Formal analysis, Supervision, Funding acquisition, Validation, Investigation, Visualization, Methodology, Writing—original draft, Writing—review and editing

### Author ORCIDs
Wannaporn Ittiprasert https://orcid.org/0000-0001-9411-8883
Shannon E Karinshak https://orcid.org/0000-0002-2079-0973
Matthew Berriman http://orcid.org/0000-0002-9581-0377
Paul J Brindley https://orcid.org/0000-0003-1765-0002

### Ethics

Animal experimentation: Mice experimentally infected with *S. mansoni*, obtained from the Biomedical Research Institute (BRI), Rockville, MD were housed at the Animal Research Facility of the George Washington University Medical School, which is accredited by the American Association for Accreditation of Laboratory Animal Care (AAALAC no. 000347) and has an Animal Welfare Assurance on file with the National Institutes of Health, Office of Laboratory Animal Welfare, OLAW assurance number A3205-01. All procedures employed were consistent with the Guide for the Care and Use of Laboratory Animals. The Institutional Animal Care and Use Committee (IACUC) of the George Washington University approved the protocol used for maintenance of mice and recovery of schistosomes. Studies with BALB/c mice involving tail vein injection of schistosome eggs and subsequent euthanasia using overdose of sodium pentobarbital was approved by the IACUC of BRI, protocol 18-04, AAALAC no. 000779 and OLAW no. A3080-01.

### Decision letter and Author response
Decision letter https://doi.org/10.7554/eLife.41337.024
Author response https://doi.org/10.7554/eLife.41337.025

## Additional files

### Supplementary files
• Supplementary file 1. Putative NHEJ and HDR pathway genes in *S. mansoni*.
DOI: https://doi.org/10.7554/eLife.41337.015

• Supplementary file 2. List of oligonucleotide primers. Nucleotide sequences and position on the schistosome ω1 gene; Smp_193860.
DOI: https://doi.org/10.7554/eLife.41337.016

• Supplementary file 3. Frequency of knock-in sequences, indels and substitutions identified in the Illumina sequencing reads, considering the entire 202 bp amplicon (except for 25 bp at the ends to exclude primer regions).
DOI: https://doi.org/10.7554/eLife.41337.017

• Transparent reporting form
DOI: https://doi.org/10.7554/eLife.41337.018

## Data availability

Database accessions Sequence reads from the amplicon NGC libraries are available at the European Nucleotide Archive, study accession number ERP110149. Additional information is available at Bioproject PRJNA415471, https://www.ncbi.nlm.nih.gov/bioproject/PRJNA415471 and GenBank accessions SRR6374209, SRR6374210.

The following datasets were generated:

| Author(s) | Year | Dataset title | Dataset URL | Database and Identifier |
|---|---|---|---|---|
| Wannaporn Ittiprasert, Victoria H Mann, Shannon E Karinshak, Avril Coghlan, Gabriel Rinaldi, Geetha Sankaranarayanan, Apisit Chaidee, Toshihiko Tanno, Chutima Kumkhaek, Pannathee Prangtaworn, Margaret M Mentink-Kane, Christina J Cochran, Patrick Driguez, Nancy Holroyd, Alan Tracey, Rutchanee Rodpai, Bart Everts, Cornelis H Hokke, Karl F Hoffmann, Matthew Berriman | 2018 | Schistosoma mansoni gene editing by CRISPR/Cas 9 targeting omega-1, exon 6 | https://www.ncbi.nlm.nih.gov/bioproject/PRJNA415471 | NCBI BioProject, PRJNA415471 |
| Ittiprasert W, Victoria H Mann, Shannon E Karinshak, Avril Coghlan, Gabriel Rinaldi, Geetha Sankaranarayanan, Apisit Chaidee, Toshihiko Tanno, Chutima KumKhaek, Pannathee Prangtaworn, Margaret M Mentink-Kane, Christina J Cochran, Patrick Driguez, Nancy Holroyd, Alan Tracey, Rutchanee Rodpai, Bart Everts | 2018 | Sequence reads from the amplicon NGC libraries | https://www.ebi.ac.uk/ena/data/view/PRJEB27995 | European Nucleotide Archive, ERP110149 |

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
