## [Decision Letter]

Thank you for submitting your article "Programmed genome editing of the omega-1 ribonuclease of the blood fluke, *Schistosoma mansoni*" for consideration by *eLife*. Your article has been reviewed by three peer reviewers, including James B Lok as the Reviewing Editor and Reviewer #1, and the evaluation has been overseen by Wendy Garrett as the Senior Editor.

The reviewers have discussed the reviews with one another and the Reviewing Editor has drafted this decision to help you prepare a revised submission.

Summary:

The three reviewers of this paper agree that it recounts a ground breaking study that warrants publication in *eLife*. We also concur that although some substantive points were raised, these can be dealt with by revising and clarifying the manuscript text and that no new experiments are required for publication.

We would characterize this paper on mutagenesis of the omega-1 (ω1) ribonuclease in *Schistosoma mansoni* to be the more comprehensive and therefore the lead of two co-submitted articles, the other being on *Opisthorchis viverrini*, which we have also recommended for publication.

With this said, the reviewers also recommend that the authors address several substantive points to achieve an improved paper. These are outlined below under Essential revisions. This paper is very well prepared and written.

Essential revisions:

The reviewers concur that revisions in four areas would substantially improve this paper.

1) The paper would be strengthened by further discussion and clarity on how to reconcile the low frequencies of mutation indicated by deep sequencing with the very large decrements in target specific mRNA and protein and in pathogenesis of mutant eggs when infused intravenously into mice. In this context, a discussion of the relative merits of the deep sequencing approach employing pools of parasites as done in the present paper versus genotyping of individual parasites if possible would be warranted. Reviewer 3 recommended this as a possible line of new experimentation, but in the interest of timely publication of this important study, our consensus is that a frank discussion of the predictive value of the deep sequencing approach and the feasibility of genotyping individual parasites will suffice.

2) Better characterization of the state of embryonic development in the *S. mansoni* eggs at the time of transduction. This concern is detailed in the substantive points raised by reviewer 3.

3) We concur that in light of the severe liver pathology that accompanies *S. mansoni* infection, reporting on any changes in severity of liver granulomas produced by mutant eggs, if these were observed, should be included. If they were not done in the study, some acknowledgement of them as a high priority for future studies would be essential.

4) The reviewers concur that this paper, and its companion on *O. viverrini*, will likely be prototypes for future studies employing CRISPR/Cas9 mutagenesis in parasitic flat worms. As such we agree that an expanded discussion of near term research priorities in solidifying programmed mutagenesis as a functional genomic tool in parasitic flatworms should be provided. What are the hurdles that remain? Specific questions to consider in this discussion would include the heritability of CRISPR-induced mutations into the F2 generation and beyond, the way forward in perpetuating lines with mutations in genes that are essential for parasite development and survival in the host and the prospects for introducing gene edits that alter functional domains in the target but do not bring about its complete knockout. Additional thoughts about this essential revision are expanded upon in the individual reviews below.

*Reviewer #1:*

This is an important manuscript, charting a significant step-forward in efforts to generate stably transformed parasitic flatworms. The manuscript is well written, all of the experiments were carefully controlled and all relevant data provided and supported through informative figures/tables – these support the main findings/conclusions.

The authors report programmed gene editing in a flatworm for the first time, charting an important advance in flatworm research broadly. The species being studied here is the most important flatworm parasite of humans, *Schistosoma mansoni*. Successful genome editing for ω1 ribonuclease was achieved using distinct methods involving electroporation with CRISPR/Cas9 or Lentiviral delivery – both knockdown and knock-in of a transgene are reported.

The phenotypic consequences of transfection of the schistosome eggs were assessed in a variety of bioassays following confirmation of reduction in target transcript and protein. Transfected fluke eggs showed reduced ribonuclease activity consistent with lower levels of target protein ω1. Further, secreted egg antigen from transfected eggs had diminished ability to drive a Th2 response. In a mouse lung model of egg-associated pathology, lungs of mice infected with transfected eggs showed dramatic reduction in circumoval granuloma formation, compared to those infected with wild type eggs.

One error spotted – in Figure 1B, exon 1 should read exon 6.

*Reviewer #2:*

The paper of Ittiprasert et al. reports on a proof of principle approach to functionally characterize ω1, an egg-specifically expressed ribonuclease of schistosomes, by CRISPR/Cas9. The authors used lentivirus and RNP-based electroporation approaches to target ω1, of which some isoforms exist in the genome. The latter impeded this pilot approach. However, the authors were able to deal with this complex scenario and, furthermore, with the fact that genetic mosaics were produced due to the fact that the eggs as biological targets for genetic manipulation are heterogeneous with respect to their stage of development ex vivo.

The authors present a comprehensive molecular and physiological study that convincingly demonstrates the possibility to successfully apply the CRISPR/Cas9 technology for functional studies also in schistosomes. Furthermore, knocking down ω1 by this approach led to phenotypes that matched to the predictions based on results of previous studies. This is groundbreaking work for the schistosome research field and in my eyes worth to be published in *eLife*.

*Reviewer #3:*

In this highly significant paper, the authors report the first proof of principle for CRISPR/Cas9 mutagenesis and editing of a target gene in a parasitic flatworm of profound medical significance. This is a groundbreaking development that will be of great interest to scientists studying parasitic flatworms. The authors have demonstrated the feasibility of inducing insertions, deletions (indels) through non-homogous end joining at a programmed site within exon 6 of the several gene copies encoding the ω1 ribonuclease in the genome of *S. mansoni*. They also demonstrated that it is possible to accomplish CRISPR/Cas9 mediated editing of a specific genomic site by providing a DNA repair template in trans that contained multiple stop codons and diagnostic PCR priming sites and was flanked by DNA arms with homology to genomic sequences flanking the predicted site of the programmed double stranded break. They provide proof that this insertion was accomplished through homology directed repair. Significantly, the authors demonstrated that eggs carrying some or all of the above mutations exhibited significant decrements in ribonuclease activity, suppressed capacities to induce production of Th2 cytokines in cultured macrophages and to incite formation of pulmonary granulomas when introduced into the systemic circulation of mice. All of these phenotypes are fully consistent with known functions of the ω1 ribonuclease of *S. mansoni* eggs in stimulating differentiation of Type 2 host macrophages and formation of egg granulomas, which are the primary lesion of human schistosomiasis. This is in exceedingly significant methodological advance in the functional genomics of schistosomes and parasitic flatworms generally.

The paper has few shortcomings but there are a few substantive aspects that the authors should consider. These are enumerated and described below. The paper is, on the whole, very well written and presented.

Substantive issues:

1) The authors employed a deep sequencing approach, CRISPResso, to detect and categorize outcomes of programmed editing of the ω1 locus in *S. mansoni* eggs. This high capacity approach was advantageous in providing a quantitative breakdown of mutations resulting from repair of double stranded breaks at the target locus by non-homologous end joining (NHEJ) or from insertion of a disrupting oligonucleotide via homology directed repair (HDR). However, results of this analysis were not straightforward to interpret, and the finding that only 4.5% of sequencing reads contained insertions, deletions or substitutions attributable to the programmed mutagenesis was particularly hard to reconcile with the >80% reduction in ω1-specific message levels in mutants, the diminution of ribonuclease activity of mutant eggs, the significantly diminished production of Th2 cytokines by macrophages in the presence of mutant eggs and the 18-fold reduction in size of pulmonary granulomas induced by these eggs. The authors provide a logical explanation for this disparity, proposing the likelihood that the electroporation and lentiviral vector approaches to transducing eggs with CRISPR elements resulted in mutagenesis of a relatively small subset of cells in the developing embryos, which were nevertheless crucial to production of ω1. This discussion would have been more satisfying if the authors, given the benefit of hindsight, had provided a more critical evaluation of this deep sequencing approach as a tool for estimating the efficiency of programmed genome editing via CRISPR/Cas9 in *S. mansoni*. Might it not be advantageous to explore the possibility of assessing the frequency of mutations in cohorts of individual parasites, either by deep sequencing of amplified genomic regions or by simple PCR detection of the novel priming site encoded in the HDR template? Would this be feasible given the amount of template that could be isolated from individual eggs? In the present circumstance, I would expect such approach to reveal that a much higher proportion of individual parasites carried the mutation than might have been predicted by the 4.5% figure derived from the CRISPResso analysis of pooled amplicons. If such a spot estimate of the proportion of transduced eggs that carry the mutation is feasible to generate for one or the other transduction protocols, this would be a strong addition to the paper, going far towards resolving the seeming paradox of rare per cell mutations resulting in strong phenotypes in pools of mutagenized eggs.

2) The paper would be improved by a more explicit description, in context, of the spectrum of development (Discussion, fourth paragraph) possible in liver eggs at the time of mutagenesis (or delivery of CRISPR elements). There are allusions in the paper to a "single celled zygote surrounded by 30 to 40 vitellin cells" and six days later "a miracidium that has developed within the egg shell". Is it possible to more accurately characterize the prevailing stage of development at the time eggs were transduced? This is important given the interpretation by the authors that mutant eggs likely represent a "mosaic of mutations". It's understood that this may be described in Dalton et al., 1997, but it would be important to clarify in the context of the present paper. In this same vein, methods for handling of the liver eggs (LE) prior to transduction with CRISPR elements should be specified in the Materials and methods (subsection “Schistosome eggs”). Most importantly, how long were LE maintained under the specified culture conditions prior to electroporation or lentiviral transduction?

3) This is a ground breaking paper. As such, I would like to have seen a concluding discussion of what the authors regard as most advantageous next steps in bringing RNA programmed mutagenesis to bear on functional genomics in schistosomes and other parasitic platyhelminths. Some questions to address might be the feasibility of establishing mutant lines of parasite, especially where the subjects of study are genes likely to be essential in negotiating molluscan or mammalian hosts. What might be prospects for regulatable mutagenesis? Given that an oligo was introduced by HDR in this study, what are the implications for introducing edits that change functional domains but do not bring about full knockout of a target gene? Are there any other salient hurdles or objectives?

---

## [Author Response]

Essential revisions:The reviewers concur that revisions in four areas would substantially improve this paper.1) The paper would be strengthened by further discussion and clarity on how to reconcile the low frequencies of mutation indicated by deep sequencing with the very large decrements in target specific mRNA and protein and in pathogenesis of mutant eggs when infused intravenously into mice. In this context, a discussion of the relative merits of the deep sequencing approach employing pools of parasites as done in the present paper versus genotyping of individual parasites if possible would be warranted. Reviewer 3 recommended this as a possible line of new experimentation, but in the interest of timely publication of this important study, our consensus is that a frank discussion of the predictive value of the deep sequencing approach and the feasibility of genotyping individual parasites will suffice.

‘To address these concerns, we have addressed the issues in an expanded Discussion section.

Specifically, the following paragraphs have been included in the revised version:

“If feasible, sequence analysis of individual schistosomes rather than pools of the parasites would provide more information, including parasite-to-parasite mutation rates. […] The ddPCR approach can provide simultaneous assessment of both homology directed repair and NHEJ, the repair pathways that resolve Cas9-catalyzed, double-stranded breaks, and also investigate multiple, simultaneous editing conditions at the target locus (Dibitetto et al., 2018, Miyaoka et al., 2018).”

and

“The possibility of large-scale deletions, as reported in *Strongyloides stercoralis* (Gang et al., 2017), presented an explanation for this apparent paradox. […] Characterization and confirmation by immunolocalization of the site of expression of ω1 in the inner envelope of the fully mature egg would be instructive.”

2) Better characterization of the state of embryonic development in the S. mansoni eggs at the time of transduction. This concern is detailed in the substantive points raised by reviewer 3.

To address this recommendation, we have included the following information in the Discussion, under the heading ‘Schistosome eggs’:

“Following mating of the adult female and male *S. mansoni*, schistosome oviposition commences at 42 days after infection of the mouse. […] The envelope of the egg, the site of expression of ω1 (Ashton et al., 2001; Mathieson and Wilson, 2010) first appears at stage 3 and is developed and secretory at the later stages (Jurberg et al., 2009).”

In addition, we revised the Materials and methods section to indicate that, following recovery from mouse livers, that the eggs (LE) were cultured in vitro for 18-24 hours before gene editing manipulation was initiated.

3) We concur that in light of the severe liver pathology that accompanies S. mansoni infection, reporting on any changes in severity of liver granulomas produced by mutant eggs, if these were observed, should be included. If they were not done in the study, some acknowledgement of them as a high priority for future studies would be essential.

We have revised the penultimate paragraph of the Discussion to address the issue, as follows:

“Given that the T2 ribonuclease ω1 is the major type 2-polarizing protein among egg-secreted proteins, these findings of a phenotype characterized by the absence of or diminutive pulmonary granulomas provide functional genomics support for this earlier advance (Evert et al., 2012). Nonetheless, given that hepatointestinal disease is characteristic of infection with *S. mansoni*, changes in severity of granulomas in the liver and intestines produced by mutant egg should be investigated as a priority in the future studies.”

4) The reviewers concur that this paper, and its companion on O. viverrini, will likely be prototypes for future studies employing CRISPR/Cas9 mutagenesis in parasitic flat worms. As such we agree that an expanded discussion of near term research priorities in solidifying programmed mutagenesis as a functional genomic tool in parasitic flatworms should be provided. What are the hurdles that remain? Specific questions to consider in this discussion would include the heritability of CRISPR-induced mutations into the F2 generation and beyond, the way forward in perpetuating lines with mutations in genes that are essential for parasite development and survival in the host and the prospects for introducing gene edits that alter functional domains in the target but do not bring about its complete knockout. Additional thoughts about this essential revision are expanded upon in the individual reviews below.

We included the paragraph in the Discussion as following:

This study provides a blueprint for editing other schistosome genes and those of parasitic platyhelminths at large, hurdles remain. […] Mutant parasite lines can be predicted to enable more comprehensive understanding of the pathobiology of these neglected tropical disease pathogens and facilitate access to novel insights and strategies for disease management.”

Reviewer #1:[…] One error spotted – in Figure 1B, exon 1 should read exon 6.

Figure 1B was corrected to read exon 6. This error has been corrected in the manuscript text.

Reviewer #3:[…] The paper has few shortcomings but there are a few substantive aspects that the authors should consider. These are enumerated and described below. The paper is, on the whole, very well written and presented.Substantive issues:1) The authors employed a deep sequencing approach, CRISPResso, to detect and categorize outcomes of programmed editing of the ω1 locus in S. mansoni eggs. This high capacity approach was advantageous in providing a quantitative breakdown of mutations resulting from repair of double stranded breaks at the target locus by non-homologous end joining (NHEJ) or from insertion of a disrupting oligonucleotide via homology directed repair (HDR). However, results of this analysis were not straightforward to interpret, and the finding that only 4.5% of sequencing reads contained insertions, deletions or substitutions attributable to the programmed mutagenesis was particularly hard to reconcile with the >80% reduction in ω1-specific message levels in mutants, the diminution of ribonuclease activity of mutant eggs, the significantly diminished production of Th2 cytokines by macrophages in the presence of mutant eggs and the 18-fold reduction in size of pulmonary granulomas induced by these eggs. The authors provide a logical explanation for this disparity, proposing the likelihood that the electroporation and lentiviral vector approaches to transducing eggs with CRISPR elements resulted in mutagenesis of a relatively small subset of cells in the developing embryos, which were nevertheless crucial to production of ω1. This discussion would have been more satisfying if the authors, given the benefit of hindsight, had provided a more critical evaluation of this deep sequencing approach as a tool for estimating the efficiency of programmed genome editing via CRISPR/Cas9 in S. mansoni. Might it not be advantageous to explore the possibility of assessing the frequency of mutations in cohorts of individual parasites, either by deep sequencing of amplified genomic regions or by simple PCR detection of the novel priming site encoded in the HDR template? Would this be feasible given the amount of template that could be isolated from individual eggs? In the present circumstance, I would expect such approach to reveal that a much higher proportion of individual parasites carried the mutation than might have been predicted by the 4.5% figure derived from the CRISPResso analysis of pooled amplicons. If such a spot estimate of the proportion of transduced eggs that carry the mutation is feasible to generate for one or the other transduction protocols, this would be a strong addition to the paper, going far towards resolving the seeming paradox of rare per cell mutations resulting in strong phenotypes in pools of mutagenized eggs.

We have discussed the novel molecular techniques to investigate indel including droplet digital PCR prior to NGS library preparation as below.

“To investigate the efficiency of programmed genome editing, several parallel approaches were undertaken including NGS- and quantitative PCR-based analysis of pooled genomic DNAs, analysis of levels of ω1*-*transcripts and ribonuclease, and immuno-phenotypic status of cultured cells and of mice exposed to gene edited eggs. […] The ddPCR approach can provide simultaneous assessment of both homology directed repair and NHEJ, the repair pathways that resolve Cas9-catalyzed, double-stranded breaks, and also investigate multiple, simultaneous editing conditions at the target locus (Dibitetto et al.,2018; Miyaoka et al.,2018).”

2) The paper would be improved by a more explicit description, in context, of the spectrum of development (Discussion, fourth paragraph) possible in liver eggs at the time of mutagenesis (or delivery of CRISPR elements). There are allusions in the paper to a "single celled zygote surrounded by 30 to 40 vitellin cells" and six days later "a miracidium that has developed within the egg shell". Is it possible to more accurately characterize the prevailing stage of development at the time eggs were transduced? This is important given the interpretation by the authors that mutant eggs likely represent a "mosaic of mutations". It's understood that this may be described in Dalton et al., 1997, but it would be important to clarify in the context of the present paper. In this same vein, methods for handling of the liver eggs (LE) prior to transduction with CRISPR elements should be specified in the Materials and methods (subsection “Schistosome eggs”). Most importantly, how long were LE maintained under the specified culture conditions prior to electroporation or lentiviral transduction?

We clarified for LE incubation time after covering from the liver, lentivirus virion exposure and DNA donor electroporation in the Materials and methods. The discussion about mosaic of mutation in parasite after gene editing also included.

“…Given the presence of multiple copies of ω1 in the genome, the numerous organs, tissues and cells comprising the mature egg, the spectrum of development of the eggs in LE (as described below in the Materials and methods), and other factors, a mosaic of mutations would be expected. Some alleles might display HDR but not NHEJ, some NHEJ but not HDR, others both NHEJ and HDR, and others retain the wild type genotype. HDR proceeds at cell division, with the cell at late S or G2 phase following DNA replication where the sister chromatid serves as the repair template; otherwise, NHEJ proceeds to repair the DSB (Heyer et al., 2010).”

3) This is a ground breaking paper. As such, I would like to have seen a concluding discussion of what the authors regard as most advantageous next steps in bringing RNA programmed mutagenesis to bear on functional genomics in schistosomes and other parasitic platyhelminths. Some questions to address might be the feasibility of establishing mutant lines of parasite, especially where the subjects of study are genes likely to be essential in negotiating molluscan or mammalian hosts. What might be prospects for regulatable mutagenesis? Given that an oligo was introduced by HDR in this study, what are the implications for introducing edits that change functional domains but do not bring about full knockout of a target gene? Are there any other salient hurdles or objectives?

We added some sentences for further study in the Discussion as below.

“This study provides a blueprint for editing other schistosome genes and those of parasitic platyhelminths at large, hurdles remain. We suggest the following (non-inclusive) list of near term priorities with respect to development of programmed gene editing for functional genomics in flatworms: the delivery of programmed genome editing to the germ line, including transgenes than confer constitutive or conditional expression of Cas9 (Idoko-Akoh et al., 2018; Chaverra-Rodriguez et al., 2018), HDR-catalyzed insertion of surrogate reporter, antibiotic resistance markers (Chaverra-Rodriguez et al., 2018) or antimetabolites to facilitate drug selection of the mutants (Yoshimi et al., 2016; Wu et al., 2014; Yan and Finnigan, 2018); and dual guide systems (Teixeira et al., 2018; Gong, 2017).